# Functional antibodies exhibit light chain coherence

David B. Jaffe[1 ✉], Payam Shahi[1,2], Bruce A. Adams[1], Ashley M. Chrisman[1], Peter M. Finnegan[1], Nandhini Raman[1], Ariel E. Royall[1], FuNien Tsai[1], Thomas Vollbrecht[1], Daniel S. Reyes[1,2], N. Lance Hepler[2,3] & Wyatt J. McDonnell[1 ✉]

The vertebrate adaptive immune system modifies the genome of individual B cells to encode antibodies that bind particular antigens[1]. In most mammals, antibodies are composed of heavy and light chains that are generated sequentially by recombination of V, D (for heavy chains), J and C gene segments. Each chain contains three complementarity-determining regions (CDR1–CDR3), which contribute to antigen specificity. Certain heavy and light chains are preferred for particular antigens[2–22]. Here we consider pairs of B cells that share the same heavy chain V gene and CDRH3 amino acid sequence and were isolated from different donors, also known as public clonotypes[23,24]. We show that for naive antibodies (those not yet adapted to antigens), the probability that they use the same light chain V gene is around 10%, whereas for memory (functional) antibodies, it is around 80%, even if only one cell per clonotype is used. This property of functional antibodies is a phenomenon that we call light chain coherence. We also observe this phenomenon when similar heavy chains recur within a donor. Thus, although naive antibodies seem to recur by chance, the recurrence of functional antibodies reveals surprising constraint and determinism in the processes of V(D)J recombination and immune selection. For most functional antibodies, the heavy chain determines the light chain.

A central challenge of immunology is the grouping of antibodies by function. Ideally, antibodies in such groups would share both an epitope and complementary paratopes dictated by their protein sequences. In practice, small numbers of antibodies are assayed in vitro–for example, for functional activities such as neutralizing capability. Larger numbers of antibodies can be assayed for simple binding to a particular antigen. In the future, antibody properties might be understood at scale from sequence information alone, perhaps via structural modelling, which could lead to antibody grouping[25]. However, in the absence of a sufficiently large dataset with multiple antigen specificities using cells from multiple humans or donors, it is currently impossible to assess the validity of any functional grouping scheme. Innovative methods such as mitochondrial lineage tracing could perhaps be used to validate computed clonotypes[26].

Nevertheless, some inferences can be made. All antibodies within a clonotype–a group of antibodies that share a common ancestral recombined cell that arose in a single donor–usually perform the same function. A clonotype can therefore be treated as the minimal functional group of antibodies. Next, as has been observed, nature repeats itself by creating similar clonotypes that appear to have the same function[2–22], and these might be combined into groups. Such recurrences have been observed between donors, but they also occur within individual donors, as we will demonstrate. Regardless, such recurrences arise after recombination randomly creates a vast pool of potential antibodies; recurrences arise through selection from that pool.

Specific examples suggest that sequence similarity can guide the way to understanding functional groups. For example, in the case of influenza virus, antibodies binding the anchor epitope of the haemagglutinin stalk domain reuse four heavy chain V genes (*IGHV3-23*, *IGHV3-30*, *IGHV3-30-3* and *IGHV3-48*) and two light chain V genes[21] (*IGKV3-11* and *IGKV3-15*). A similar observation has been made in the case of Zika virus, where a protective heavy–light chain gene pair *IGVH3-23–IGVK1-5* is observed in multiple humans, that also cross-reacts with dengue virus[16]. Even in the setting of HIV infections, which lead to diverse and divergent viruses within a single human, recurrent and ultra-broad neutralizing antibodies such as the VRC01 lineage emerge, with subclass members using combinations of heavy chains encoded by *IGHV1-2* and *IGHV1-46* paired with light chains encoded by *IGKV1-5*, *IGKV1-33*, *IGKV3-15* and *IGKV3-20*[22].

Motivated by these examples, we set out to determine whether unrelated B cells with similar heavy chains also have similar light chains. We exclude related cells (that is, those in the same clonotype) because they use the same VDJ genes by definition.

We generated a large set of paired V(D)J data to investigate this question. Using peripheral blood samples from four unrelated humans (Methods and Extended Data Table 1), we captured and sequenced paired, full-length antibody sequences from a total of 1.6 million single B cells of 4 phenotypes defined by flow cytometry[27,28]: naive, unswitched memory, class-switched memory and plasmablasts (Extended Data Fig. 1). For each cell, we obtained nucleotide sequences spanning from

[1]10x Genomics, Pleasanton, CA, USA. [2]These authors contributed equally: Payam Shahi, Daniel S. Reyes, N. Lance Hepler. [3]Unaffiliated: N. Lance Hepler. ✉e-mail: 99.david.b.jaffe@gmail.com; wyattmcdonnell@gmail.com

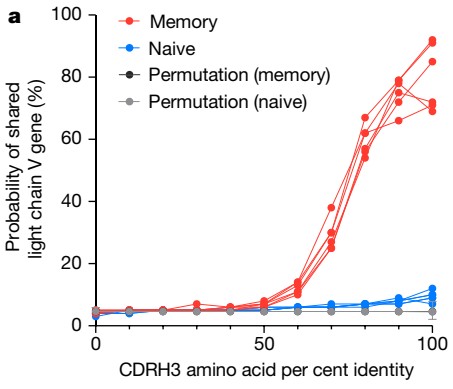

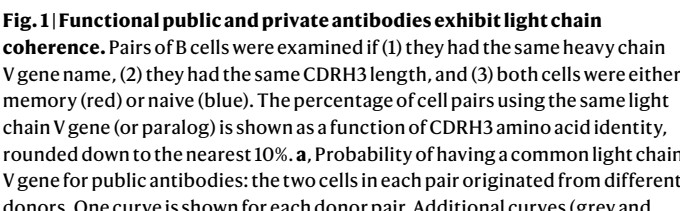

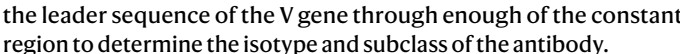

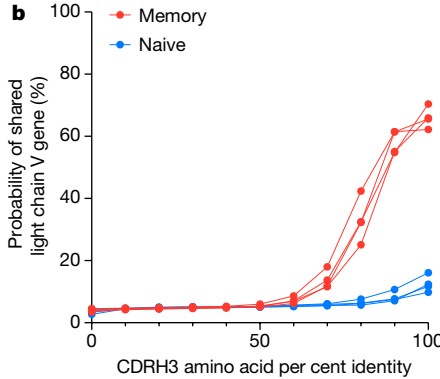

**Fig. 1 | Functional public and private antibodies exhibit light chain coherence.** Pairs of B cells were examined if (1) they had the same heavy chain V gene name, (2) they had the same CDRH3 length, and (3) both cells were either memory (red) or naive (blue). The percentage of cell pairs using the same light chain V gene (or paralog) is shown as a function of CDRH3 amino acid identity, rounded down to the nearest 10%. **a**, Probability of having a common light chain V gene for public antibodies: the two cells in each pair originated from different donors. One curve is shown for each donor pair. Additional curves (grey and

black (hidden below the grey in the graph)) show the light chain coherence when heavy and light chain correspondence is randomly permuted. Data are mean ± s.e.m. We tested the differences between regression curve slopes using a sum-of-squares $F$ test ($P < 0.0001$, $F = 20.89$, d.f. numerator = 13, d.f. denominator = 140). **b**, Probability of having a common light chain V gene for private antibodies: the two cells in each pair are from the same donor, but from different computed clonotypes and exhibit additional evidence that they lie in different true clonotypes (Methods). One curve is shown for each donor.

the leader sequence of the V gene through enough of the constant region to determine the isotype and subclass of the antibody.

We computationally split these antibody sequences into two types: naive and memory. To do so, we inferred V gene alleles for each of the four donors (Methods), and then for each B cell used the inferred alleles to estimate the number of somatic hypermutations (SHMs) that occurred outside the junction regions, including both chains. We labelled an antibody sequence as naive if it had no mutations relative to the inferred germline (that is, exhibited no SHM), and as memory otherwise, understanding that these categories are biological oversimplifications and do not account, for example, for class switching. We compared these categories with the flow-sorted categories (Extended Data Table 2). Approximately 80.0% of cells sorted as naive were naive by computational analysis, with a maximum of 90.9% for donor 2. Conversely, just 0.6% of cells sorted as memory were naive by computational analysis. During library preparation, we exhausted the supply of memory cells and deliberately mixed them in some libraries (for example, switched B cells plus naive B cells) to best exploit capacity. Our computational sorting also enabled us to make the best use of all the data.

## Memory antibodies show light chain coherence

Next, we investigated whether for unrelated B cells, similar heavy chains imply similar light chains. We explored this question separately in memory and naive cells by considering pairs of cells, either both memory or both naive. We considered only pairs of cells with the same heavy chain V gene and the same CDRH3 length, and whose cells came from different donors. We divided the pairs into 11 sets according to their CDRH3 amino acid per cent identity, rounded down to the nearest 10%. Then for each set, we computed its light chain coherence: the percentage of cell pairs in which the light chain gene names were identical. In this work, we consider light chain V gene paralogs with D in their name to be identical (for example, *IGKV1-17* and *IGKV1D-17* are considered identical), as previously described[29].

We show the results of this analysis in Fig. 1a. For memory B cells found in separate donors having the same heavy chain V genes and 100% CDRH3 amino acid identity (2,813 cells), we found 82% coherence between their light chains, whereas light chain coherence in naive cells (754 cells) was only 10%. This makes sense, as naive cells have generally not yet been selected for functionality or undergone SHM during an immune response. This finding implies that for memory cells, which

bear functional antibodies and are typically the products of thymic and peripheral selection, heavy chain coherence implies light chain coherence. We further tested light chain coherence for memory B cells using independent datasets, finding 93% coherence on a heterogeneous compendium of older data (data generated by 10x Genomics for internal development and distributed with enclone[30], as well as several other datasets related to multiple sclerosis[31], COVID-19[32,33], Kawasaki disease[34] and LIBRA-seq[35]) and 79% coherence on a recent dataset[36] (Methods, 'Additional data'). We redistribute these data as part of our paper and provide details about each dataset in Supplementary Table 1.

We also analysed these data using only one cell per clonotype, finding 79% light chain coherence for memory B cells for the data in this work (Methods and Supplementary Table 2). Moreover, to completely eliminate the possibility that light chain coherence might be explained by cross-sample contamination, we compared pairwise between our data, the older data, and the recent data, treating each as a single super-donor, finding light chain coherence of 70%. We tested the effect of cell misclassification by randomly swapping memory and naive labels for 10% of cells, finding 82% coherence for memory cells and 17% (versus 10% without swapping) coherence for naive cells, suggesting that naive coherence might be exaggerated by actual labelling errors. We note that light chain coherence does not imply heavy chain coherence (Extended Data Table 3). We also note that if we instead define naive (CD19+IgD+CD27±CD38±CD24±) and memory cells (Methods) by flow cytometry, we found 86% concordance between light chains for memory cells (87.3% for switched and 84.4% for unswitched) and 16% for naive cells. Supplementary Figs. 1 and 2 show significant associations between VH and VL genes and superfamilies and the frequency of VL superfamily usage by CDRH3 length.

Light chain coherence remains even if light chain V gene paralogs are not treated as the same gene, with light chain coherence of 64% for memory cells (Extended Data Table 4). A more sophisticated approach might make more identifications, as sufficiently similar V genes should be functionally indistinguishable. In fact, light chain coherence can be observed without reference to genes at all. Given pairs of cells from different donors, we can compute their heavy and light chain edit distances. In that case, if the cells have the same CDRH3 amino acid sequence, 78% of the time their light chain edit distance is less than or equal to 20, whereas without the CDRH3 restriction this is true only 9% of the time (Extended Data Fig. 2).

Recurrences (separate recombination events) occur between different donors and also within single donors; from first principles, light

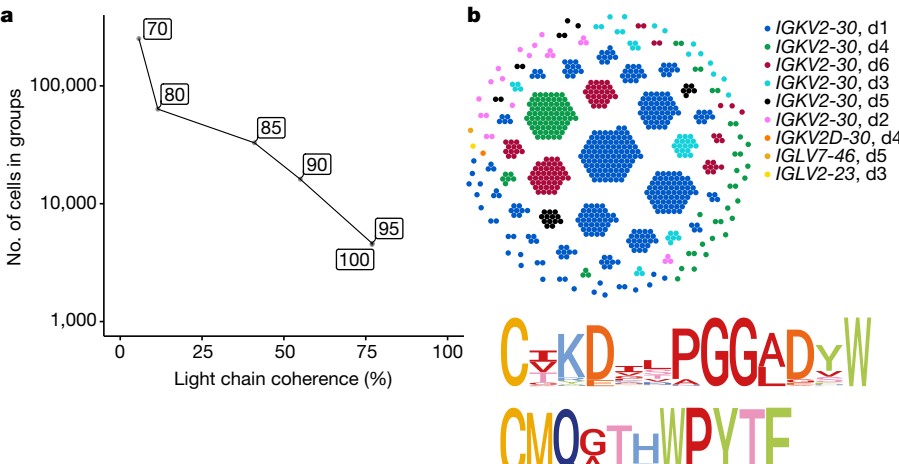

**Fig. 2 | Transitive linking yields large coherent groups. a**, We transitively grouped clonotypes at given per cent CDRH3 amino acid identities (boxed) while requiring the same heavy chain V gene. The graph shows the relationship between light chain coherence and the number of cells appearing in each group. **b**, Top, at 90% identity, the transitive group containing a cell with the CDRH3 sequence CIKDILPGGADSW is shown, using the data from this Article and from Phad et al. (2022)[36]. These cells use the heavy chain V gene *IGHV3-9*. Each dot represents a cell, and each cluster is a computed clonotype. With the exception of three cells, all computed clonotypes use the light chain gene *IGKV2-30*, and cells from all six donors (d1, d2, d3, d4, d5 and d6) are present. Bottom, logo plot for CDRH3 (top) and CDRL3 (bottom) amino acid sequences in this group.

chain coherence are expected to occur at a similar rate within a single donor. This is difficult to investigate, because SHM could cause two cells from the same recombination event to manifest as separate events. We addressed this by considering cells in different computed clonotypes, which—if the computation were perfect—would have arisen from independent recombinations. We required additional conditions to further increase the likelihood that the cells in fact represented a true recurrence (Methods). For example, we treated it as sufficient for two cells to have different CDRL3 lengths, as this would in all likelihood arise from separate recombination events—although this cannot be confirmed using VDJ sequencing alone. Using this approach, we observed 65% light chain coherence for memory cells at 100% CDRH3 amino acid coherence for cells from the same donor (Fig. 1b). This is lower than observed for cells from different donors (Fig. 1a), probably because the additional precautionary conditions (Methods) remove many cases of bona fide recurrent antigen-specific responses that would be expected to arise within a human.

As an analytical tool, pairs of cells are useful for understanding VDJ biology. However, it is the non-overlapping groups of memory B cells that might share function that are of greatest interest biologically. The following grouping scheme is an example of a common approach to this problem: placing all memory B cells sharing identical heavy chain V genes and 100% identical CDRH3 amino acid sequences in the same group. This is unsatisfactory, because it ignores clonotypes that share function but differ in amino acid sequence, and because some amino acids can change without affecting the ability of an antibody to bind its antigen.

These limitations are readily overcome. We provide a concrete example at 90% identity, for the sake of clarity. We consider only memory cells and only computed clonotypes consisting entirely of memory cells. We define groups by first defining when two cells are similar. We call two cells similar if they belong to the same clonotype or if they have identical heavy chain V genes and 90% identical CDRH3 amino acid sequences. We then place two cells X and Y in the same group if there is a sequence of cells $X = X_1, \ldots, X_n = Y$ such that for each $i$, $X_i$ is similar to $X_{i+1}$. The multiple 'hops' between these two cells make them transitively similar. This process is a well-known mathematical notion, which we call transitive grouping. It places every cell in a group and yields non-overlapping groups. We analysed these groups for light chain coherence by examining pairs of cells from different donors within the same group.

Transitivity enables the formation of large groups of clonotypes. As Fig. 2a shows, the light chain coherence of these groups decreases as the CDRH3 per cent identity is lowered. This is more rapid than might be expected because of the multiple hops in transitivity, each of which has the potential to connect antibodies having different functions. Even so, large coherent groups can be formed. Figure 2b shows a transitive group of clonotypes computed using 90% percent identity, using both the data of this work and that of Phad et al.[36] and sharing the heavy chain V gene *IGHV3-9*. The light chain coherence for this group is 99.7% (726 out of 728 cells), and all but four cells use the light chain genes *IGKV2-30* and *IGKJ2*. We note that of memory B cells using the heavy chain V gene *IGHV3-9* in these datasets, only 7% (1,203 out of 16,880) use *IGKV2-30*. The cells in the group lie in 122 computed clonotypes, each of which would represent an independent recurrence (recombination) event if the clonotype computation were perfect. Both heavy and light CDR3 sequences exhibit strong conservation. Notably, 93% of the cells have subclasses IgA1 or IgA2.

## Simulation predicts antibody recurrence

Antibody recurrences would be expected to arise preferentially from relatively common recombination events. We analysed each junction sequence by finding the most likely D region, allowing for no D region or a concatenation of two D regions to account for VDDJ junctions[37,38], and aligned the antibody nucleotide sequence to the concatenation of the V(D)J reference sequences (Fig. 3a and Tables 1 and 2). We first tested whether the observed recurrence rate was comparable to that expected by chance. Answering this poses a dilemma as it requires deep enough knowledge of recombination to accurately recapitulate the process by simulation. Other researchers have addressed this challenging problem[39–41]. We generated random antibody junction sequences using the simulation program soNNia[41] using deduplicated, naive heavy chain nucleotide sequences from our data for training. We also explored simulation variants (see Methods and Supplementary Table 3). Our simulations leverage two features that were not used in previous studies[23,24]. Namely, we identify and use for simulator training truly naive junction sequences (with no detectable SHM in the other CDR or FR regions), and we account for central and peripheral selection using the post-selection model[41].

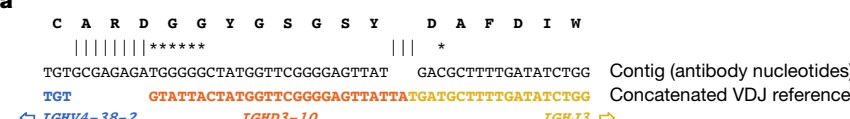

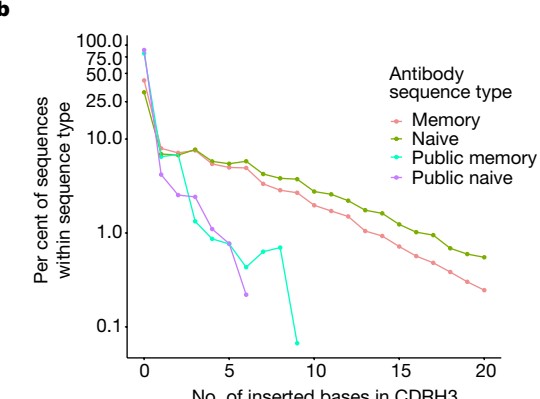

**Fig. 3 | Public antibody properties are consistent with recombination biology.** Heavy chain junction sequences were aligned to concatenated reference sequences comprising VJ, VDJ or VDDJ, with up to two different D genes, and the most likely reference. We determined the number of bases inserted in the junction relative to this reference (counting deletions separately), as well as the number of substituted bases. **a**, The heavy chain junction region for a memory cell with the heavy chain junction CARDGGYGSGSYDAFDIW is shown. We found *IGHD3-10* to be the most likely D gene. There are eight inserted bases in the junction and seven substitutions. The substitution rate is 7 out of 46, where the denominator (46) is the total number of matching and mismatching bases. **b**, For each of four types of antibodies in the data, we computed the number of inserted bases in the heavy chain junction region, relative to the concatenated VDJ (or in some cases VJ or VDDJ) reference sequence. Most of the inserted bases are insertions in non-templated region 1 or 2. The frequency is shown as a function of the number of inserted bases. See also Table 1.

We then calculated how many recurrent naive antibodies are predicted by simulation, by making four groups of simulated antibodies of the same sizes as the groups of naive cells in our data, and then counting cross-donor recurrences of heavy chain gene and CDRH3 amino acid pairs from each of ten simulation replicates. Recurrences of naive antibodies are exhibited as cell counts, which makes sense because naive cells rarely appear in clonotypes having more than one cell. Whereas the actual number of recurrences we observed was 754, the average predicted value was 1,190. Given the challenges of simulation, these numbers are close. This confirmed that the soNNia simulator recapitulates statistical properties of real repertoires, including the frequency of VDDJ recombinations (Table 1) and the recurrence of junctions even when training the model on memory rather than naive sequences (Supplementary Table 3).

Next, we investigated whether the junction regions of recurrent antibodies are as complex as those of arbitrary antibodies (Fig. 3b). We found that recurrent antibodies have an order of magnitude fewer inserted bases, and that this is true for both naive and memory cells. This shows that recurrent antibodies have intrinsically less complex junctions than arbitrary antibodies and—as expected—are thus more likely to recur by chance. In fact, most recurrent antibodies, whether naive or memory, have no inserted bases (Fig. 3b).

Finally, we investigated light chain coherence in antibodies that recur in a few individuals. On average, such antibodies have relatively simple junction regions (Fig. 3b). Antibodies with more complex junctions should recur at a lower rate. Within a larger population of individuals these would be highly visible, but they can still be observed in our data. We found that recurrent antibodies with more inserted nucleotides (non-templated regions 1 and 2, and other inserted bases relative to the reference sequence of the junction alignment) have greater light chain coherence than those with no inserted nucleotides (Table 2). This lessened our concern that recurrent antibodies are exceptional with respect to light chain coherence. Indeed, our findings suggest that all antibodies are recurrent, but at varying rates depending on their junction complexity and the prevalence of their cognate antigen. Our findings also suggest that aside from frequency, more complex antibodies do not behave differently with respect to light chain coherence.

## Discussion

This work supports the following model, generalizing observed constraints on gene usage by some antibodies[2–24]. In nature, many heavy chain configurations yield effective binding of a given antibody target. However, for each of those heavy chain configurations, the cognate light chain is largely determined—at a rate of around 80% at the level of light chain gene or paralog. We call this phenomenon light chain coherence and observe it by looking for recurrences of heavy chains in

## Table 1 | Public antibody properties are consistent with recombination biology

| Data source | Recurrences | | Average junction property | | | | |
|---|---|---|---|---|---|---|---|
| | All cells (mean±s.e.m.) | VDDJ cells (mean±s.e.m.) | Insertion length (nt) | Substitution rate in junctions from | | VDDJ rate | CDRH3 length (amino acids) |
| | | | | All cells | Cells with no insertions | | |
| Real | 754.0 | 0.0 | 5.0 | 15.6% | 7.2% | 0.52% | 18.4 |
| Simulated | 1,189.6 ± 10.8 | 1.8 ± 0.7 | 5.3 | 16.5% | 7.3% | 0.32% | 18.1 |

Junction properties of observed and simulated antibody sequences in this study, using ten simulation replicates (see also Fig. 3 and details of simulations in Methods). nt, nucleotides.

**Table 2 | Light chain coherence in complex antibody junctions**

| No. of inserted bases in junction | Minimum CDRH3 amino acid identity | | | | | |
|---|---|---|---|---|---|---|
| | 100% | | 90% | | 80% | |
| | Cell pairs | LCC (%) | Cell pairs | LCC (%) | Cell pairs | LCC (%) |
| 0 | 4,307 | 80.1 | 18,123 | 71.5 | 109,751 | 58.0 |
| 1 | 188 | 97.3 | 1,616 | 95.0 | 10,044 | 75.3 |
| 2 | 134 | 84.3 | 1,626 | 86.3 | 16,704 | 85.4 |
| 3 | 21 | 85.7 | 449 | 65.0 | 4,808 | 54.2 |
| 4 | 7 | 100.0 | 74 | 89.2 | 1,839 | 63.5 |
| 5 | 0 | | 34 | 82.4 | 1,012 | 71.2 |
| 6 | 0 | | 64 | 81.2 | 1,176 | 68.8 |
| 7 | 0 | | 60 | 95.0 | 826 | 82.2 |
| ≥ 8 | 18 | 100.0 | 80 | 90.0 | 656 | 67.5 |

For pairs of memory cells as in Fig. 1, using varying CDRH3 amino acid identity we show the light chain coherence (LCC) as a function of the number of inserted bases in the junction sequence (see also Fig. 3).

memory and naive cells from four donors. The small number of donors biases our analysis towards junction regions with low complexity, although the same phenomenon occurs for more complex junctions that appear in our data. Our findings suggest that light chain coherence may apply to memory B cells in general.

Although we analysed V(D)J data for only around 2 million cells in this work, deeper data have been generated separately for heavy and light chains. This has enabled the identification of recurrences (public clonotypes) within such data, using strict definitions based on 100% CDRH3 amino acid identity[23,24]. We reinterpret these data here with caution, owing to the differences in scale and technical approaches between the studies. We show here that previously described recurrences in these and other studies come from two types of B cells: naive B cells making 'not yet functional' antibodies with minimal light chain coherence and memory B cells making functional antibodies with light chain coherence. The lack of light chain coherence in naive B cells is expected, given their lack of acquired functionality and selection. Conversely, the light chain coherence in memory B cells is expected because of their acquired functionality and selection.

The simplest explanation for the recurrence that we and others observe between naive cells is that their sequences repeat purely by chance. We show that in our data, recurrent naive cells (as well as memory cells) have markedly lower junction complexity, and it is thus no surprise that they recur. One might ask whether the observed recurrence frequency is consistent with the mechanistic biology of V(D)J recombination. Answering this question would require precise quantitative knowledge regarding this exquisitely complex process, and such knowledge does not exist. However, we show here that simulation of this process does in fact predict recurrences, at a similar rate to our observations. Thus, we propose that naive sequences recur by chance. Conversely, recurrent memory sequences are a product of both chance and common exposure to related antigens. We suggest that recurrent naive sequences are instructive with respect to recombination and that recurrent memory sequences are instructive with respect to antibody function, central selection and peripheral selection.

We postulate that light chain coherence implies functional coherence. However, we do not claim to solve the problem of functional grouping. For this to be possible, at least two hurdles remain. First, all approaches (including ours) that are based on direct sequence comparison are naive to the structural consequences of amino acid changes. Rather than compare sequences, a more effective route to functional grouping

may be to first computationally model antibody structures from their sequences and to then compare those structures[25,42–45]. Second, far better truth data are needed to assess any method. It follows that although similar antibodies for the same antigen have been widely observed in multiple individuals, it remains unknown how frequently antibodies to different antigens might be equally similar. Truth data targeted at such questions could comprise a suite of large datasets of naturally occurring antibodies, with one dataset for each of several antigens, along with binding data for each antibody. These data would be most powerful if they included nucleotide sequences (enabling, for example, consistent VDJ gene identification) and enough donor information to distinguish bona fide recurrence from clonal expansion within given individuals. The generation of such truth data at scale is feasible using existing methods[21,35,46–48]. The immunoglobulin loci and their products are complex and challenging to analyse. It is reasonable to assume that some sequences treated as naive in this study are in fact antigen-specific, as others have described in SARS-CoV-2 infection[49]. Conversely, although progress has been made in the identification of novel germline alleles[50,51], some sequences treated as memory in this study may in fact be naive and produced by novel alleles not detected by our inference methods.

By virtue of how V(D)J recombination works, the light chain sequences of antibodies carry less information than the heavy chain sequences[1]. Our work reveals that the light chains of functional antibodies are highly constrained, which implies that the light chains used in nature are the most functional ones: natural selection has won out. The complex dance between the heavy chain and the light chain is best studied at the level of individual cells, the context in which antibodies are produced, selected and expanded. Although we do not yet understand why, we show that the choreography of this interaction leads to a limited number of acceptable light chains. We suggest that antibody designers would be wise to actively look for optimal light chains used widely in nature, rather than focusing on the heavy chain alone. Similarly for bispecific antibodies, it could be advantageous to find two heavy chains whose native light chains are similar.

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

## Methods

For several details, we refer to ref. [51]. We provide commands using executables in the enclone code and data of this work at https://github.com/DavidBJaffe/enclone/blob/master/enclone_paper/which_code_does_what. A single line can be used to install the enclone executable and download the data (https://10xgenomics.github.io/enclone/). We used version 0.5.175 of enclone. The enclone runs on the data of this work require about 145 GB of memory and 20–40 min on a multi-core server (24 cores). As an intermediate, we generated a file per_cell_stuff (https://plus.figshare.com/articles/dataset/Dataset_supporting_Functional_antibodies_exhibit_light_chain_coherence_/20338177?file=36366549) with one line per cell, to facilitate direct analysis of the data of this work by other methods. There are also files per_cell_stuff.old_data (https://plus.figshare.com/articles/dataset/Dataset_supporting_Functional_antibodies_exhibit_light_chain_coherence_/20338177?file=36366552) and per_cell_stuff.phad (https://plus.figshare.com/articles/dataset/Dataset_supporting_Functional_antibodies_exhibit_light_chain_coherence_/20338177?file=36366543) corresponding to the other data that were used. Use of the other executables requires compilation from source code available at https://github.com/DavidBJaffe/enclone, and are in its directory enclone_paper/src/bin. These calculations (typically taking as input the files such as per_cell_stuff) were run on a MacBook Pro with 16 GB of memory.

### Flow cytometry

We used a Sony MA900 cell sorter to purify single B cell suspensions from PBMCs from the four donors described in this paper. We used the following flow gating definitions for each population: naive: live, CD3−CD19+IgD+CD27±CD38±CD24±; unswitched memory: live, CD3−CD19+CD27+IgDlowIgM++CD38±CD24±; switched memory: live, CD3−CD19+CD27+IgD−CD38±CD24±CD95±; and plasmablast: live, CD3−CD19+CD27+IgD−CD38++CD24−.

The antibody panel we used comprised of the following eight clones in a stain volume of 200 μl, totalling 1.25 μg of antibody with each antibody at a 1:40 dilution (5 μl): LIVE/DEAD dye (Invitrogen, 7-AAD, 00-6993-50), anti-CD3 (BioLegend, BV711, 317327, clone OKT3, IgG2a/kappa, mouse), anti-CD19 (BioLegend, PE, 982402, clone HIB19, IgG1/kappa, mouse), anti-IgD (BioLegend, APC, 348221, clone IA6-2, IgG2a/kappa, mouse), anti-IgM (BioLegend, PE/Dazzle 594, 314529, clone MHM-88, IgG1/kappa, mouse), anti-CD24 (BioLegend, BV605, 311123, clone ML5, IgG2a/kappa, mouse), anti-CD27 (BioLegend, FITC, 302805, clone O323, IgG1/kappa, mouse), anti-CD38 (BioLegend, BV421, 356617, clone HB-7, IgG1/kappa, mouse), and anti-CD95 (BioLegend, BV510, 305639, clone DX2, IgG1/kappa, mouse).

We titrated and developed the B cell fractionation panel using 20 million fresh PBMCs (AllCells, 3050363) from a healthy human donor whose cells were not used to generate single-cell data in this study. We thawed the cells per the 10x Genomics Demonstrated Protocol for Fresh Frozen Human Peripheral Blood Mononuclear Cells for Single Cell RNA Sequencing (CG00039, Revision D). In brief, we resuspended cells in 20 μl PBS/2% FBS and incubated the cells on ice for 30 min in the dark. Before sorting, we washed the cells in 3× 1 ml PBS/2% FBS and then resuspended them in 300 μl PBS/2% FBS for the sort step.

### Single-cell data generation

Cells from four donors were flow sorted as naive, switched memory, unswitched memory, and plasmablast. Donor blood was collected under IRB-approved protocols with informed consent managed by AllCells (a subsidiary of Discovery Life Sciences); no clinical trials were conducted and no personally identifying information is reported in this study or its data. V(D)J sequences were obtained using the 10x Genomics Immune Profiling Platform, using six Chromium X HT chips and standard manufacturer methods. cDNA libraries were sequenced on the NovaSeq 6000 platform on S4 flow cells. Certain memory

B cell populations were relatively uncommon within certain donors. To account for this, in some libraries we added naive B cells to isolated memory cells from each donor in order to capture a sufficiently large number of total B cells and unique sequences from each donor (see also Extended Data Table 1). We targeted 20,000 cells recovered from each lane on the HT chip.

### Additional data

We used several other datasets. These included single-cell data for 247,516 cells from Phad et al.[36], which were approximately evenly distributed between two donors. A second combined collection ('older data') included data generated by 10x Genomics for internal development, and distributed with enclone, as well as several other datasets related to multiple sclerosis[31], COVID-19[32,33], Kawasaki disease[34] and LIBRA-seq[35].

Some of these data predated dual indexing and showed evidence of mixing (that is, index hopping) within a given flow cell. In such cases we treated cells from multiple donors as belonging to a single donor. After merging there were 23 donors, and a total of 280,669 cells in the older data.

### Additional condition on pairs of cells from the same donor

Pairs of cells were chosen from different computed clonotypes (Fig. 1b). In order to make it very unlikely that the two cells belonged to the same true clonotype, we required in addition that at least one of three conditions was satisfied: (1) The data support different heavy chain J gene usage for the two cells. For this, we examined framework region four for both cells. There had to exist at least three positions where the reference sequences were different, and the cells had bases consistent with those, and no positions where the reference sequences were different, and the cells both supported just one of the references. (2) The same criteria as in (1) but for the light chain instead of the heavy chain. (3) The light chain CDR3 lengths differed.

### Sequence simulation

We used soNNia v0.1.2, commit 85c7169, to simulate 10 replicates of 1,408,939 heavy chain junctions. A reproducible Conda environment, scripts to generate these files, and simulation data are provided at https://plus.figshare.com/articles/dataset/Dataset_supporting_Functional_antibodies_exhibit_light_chain_coherence_/20338177?file=36354231. We trained two soNNia models using naive-annotated sequences in pre-selection and post-selection mode, and two models using memory-annotated sequences in pre-selection and post-selection mode.

### Statistical testing

**Permutation analysis of light chain coherence.** We performed a permutation test of light chain coherence by permuting light chain data, while leaving heavy chain data fixed, then calculating light chain coherence between pairs of cells as in Fig. 1, using only one cell per clonotype to reduce potential bias from clonal expansions. We performed 1,000 permutations at values of light chain coherence between 0% and 100% by steps of 10% and then calculated the standard error of the mean at each level. We show these curves in Fig. 1a. We tested for difference between slope coefficients of linear regression models of these curves using a sum-of-squares $F$ test.

**VH/VL contingency test.** We tested the significance of the VH/VL contingency table (where each count is a clonotype with a given VH/VL pair) using Monte Carlo simulation of 100,000 $P$ values.

### Sequence logo plots

We used the ggseqlogo[52] and msa[53] R packages to align CDRH3 and CDRL3 sequences with the MUSCLE algorithm and to generate logo

plots using position-wise entropy/Shannon information ($y$ axis unit: bits). Letters were coloured based on the properties of various amino acids. The amino acid-property colour coding and other code necessary to reproduce these figures are publicly available as part of this paper.

### Allele inference

Donor alleles for V genes are partially inferred (as part of the enclone software, in the file allele.rs), using the following algorithm. The core concept is to pile up the observed sequences for a given V gene and identify variant bases. If we used all cells (one sequence per cell), then the sequences would be biased by clonal expansion and thus yield incorrect alleles. Ideally we would instead use just one cell per clonotype that uses the given V gene. However the order of operations is that we first compute donor alleles, and then compute clonotypes. Therefore we use a heuristic for picking cells that does not depend on knowing the clonotypes. The heuristic is that we pick just one cell among those using the given V gene, and that share the same CDRH3 length, CDRL3 length, and partner chain V and J genes. The pileup is then made from the V gene sequences of these cells.

Next, for each position along the V gene, excluding the last 15 bases (to avoid the junction region), we determine the distribution of bases that occur within these selected cells. We only consider those positions where a non-reference base occurs at least four times and represents at least 25% of the total. Then each cell has a footprint relative to these positions, which is its list of base calls for the given positions; we require that these footprints satisfy similar evidence criteria. Each such non-reference footprint then defines an 'alternate allele'. We do not restrict the number of alternate alleles because they could arise from duplicated gene copies. The ability of the algorithm to reconstruct alleles is limited by the depth of coverage (counted in 'non-redundant' cells) of a given V gene. Moreover the algorithm cannot identify germline mutations which occur in the terminal bases of the V gene, inside the junction region. An example of allele calling may be found in ref. [51], in the section on donor reference analysis.

### Reporting summary

Further information on research design is available in the Nature Research Reporting Summary linked to this article.

### Data availability

All data are publicly available at https://doi.org/10.25452/figshare.plus.20338177, including processed full-length V(D)J sequences and annotations.

### Code availability

All code to replicate key findings and figures of the paper are available at https://github.com/DavidBJaffe/enclone (Git hash 561e3ac); a separate copy of this code has also been deposited on Figshare+ at https://plus.figshare.com/articles/dataset/Dataset_supporting_Functional_antibodies_exhibit_light_chain_coherence_/20338177?file=37819143.

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

**Acknowledgements** We wish to express our gratitude to the donors for their patience in undergoing apheresis and donating their blood. We thank P. Marks, M. Stubbington, S. Taylor, P. Shah and V. Kumar for their comments and suggestions. Funding and resources were provided by 10x Genomics.

**Author contributions** D.B.J. and W.J.M. were responsible for conceptualization, formal analysis, methodology, software, supervision, validation and writing of the original draft manuscript. D.B.J., P.S., B.A.A., N.R., D.S.R., N.L.H. and W.J.M. were responsible for data curation. All authors were responsible for investigation. D.B.J., P.S., D.S.R., N.L.H. and W.J.M. were responsible for project administration. D.B.J., P.S., N.R. and W.J.M. were responsible for visualization. All authors were responsible for review and editing of the final draft manuscript.

**Competing interests** All authors except N.L.H. were employees of 10x Genomics at the time of submission. Several authors were also shareholders of 10x Genomics at the time of submission. D.B.J., P.S., B.A.A. and W.J.M. are inventors on patent applications assigned to 10x Genomics in relation to algorithms and methods for the study of immune repertoires.

**Additional information**
**Correspondence and requests for materials** should be addressed to David B. Jaffe or Wyatt J. McDonnell.

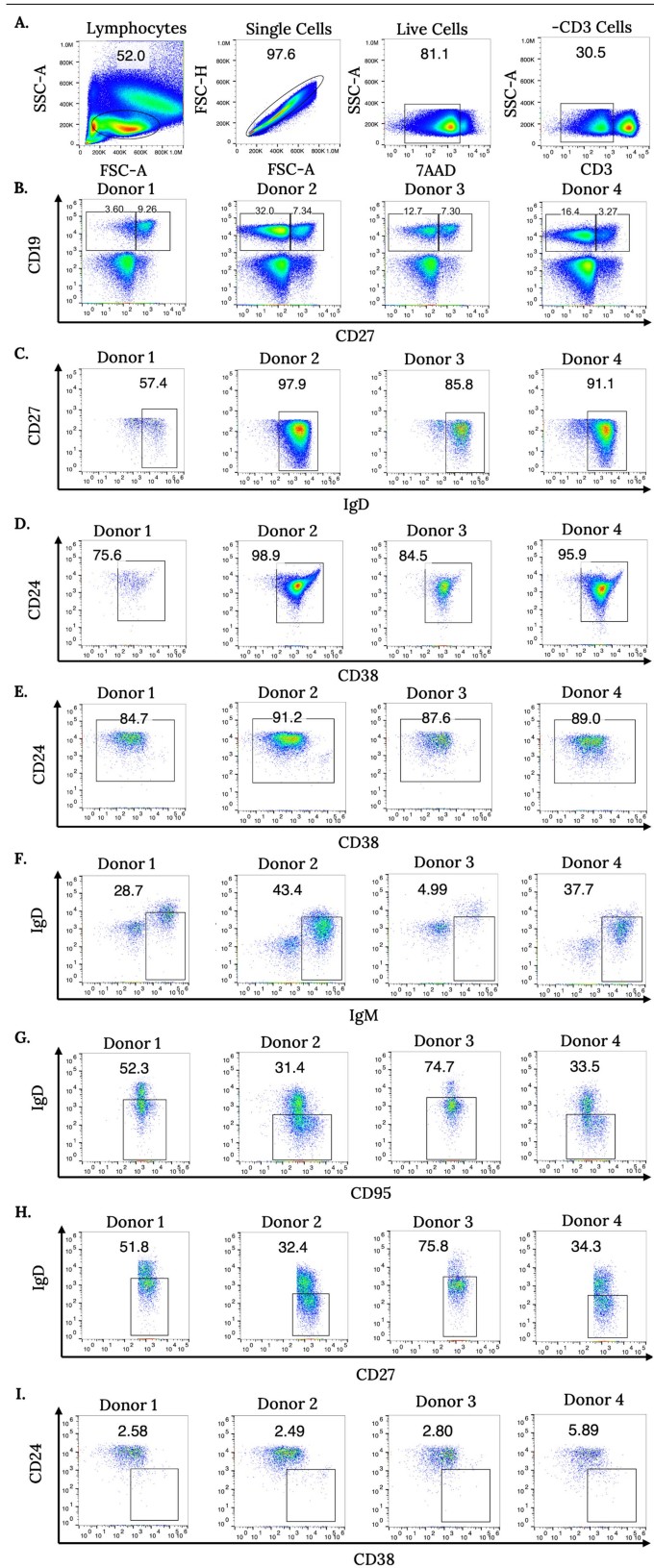

**Extended Data Fig. 1** | See next page for caption.

**Extended Data Fig. 1 | Flow cytometry gating schemes for B cell subsets.**
Gating strategy for B cell isolation. Panels **a**–**d** show naive cell gating, panels **e**–**g** show memory cell gating, and panels **h**–**i** show plasmablast gating. **a**, Hierarchical gating scheme for lymphocytes, single cells, live cells, and CD3-negative cells. **b**, We gated CD19+CD27± cells from CD3− cells for further analysis. Donor samples displayed noticeable differences in CD19 and CD27 expression. **c**, We analyzed CD19+CD27− cells for surface IgD expression and gated IgD+ cells for further analysis. **d**, We selected naive B cells by sorting CD19+CD27−IgD+CD24±CD38± B cells. **e**, For memory cell gating, we selected CD19+CD27+ cells from **b** for CD24+CD38+ positivity. **f**, We analyzed cells from **e** and isolated unswitched memory cells using IgD±IgM++ gating. **g**, We analyzed cells from **e** and isolated switched memory cells using IgD−CD95+ gating. **h**, We analyzed CD19+CD27+ cells from **b** and gated the IgD−CD27+ population. **i**, We sorted plasmablasts using CD24−CD38++ gating.

**(a)**

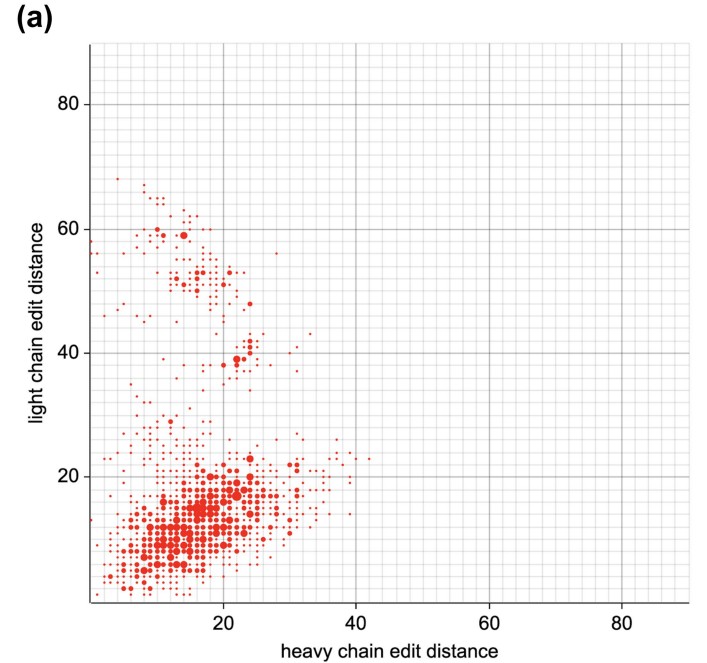

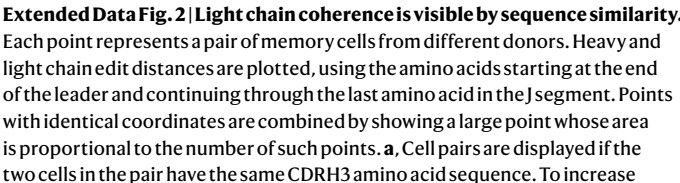

**(b)**

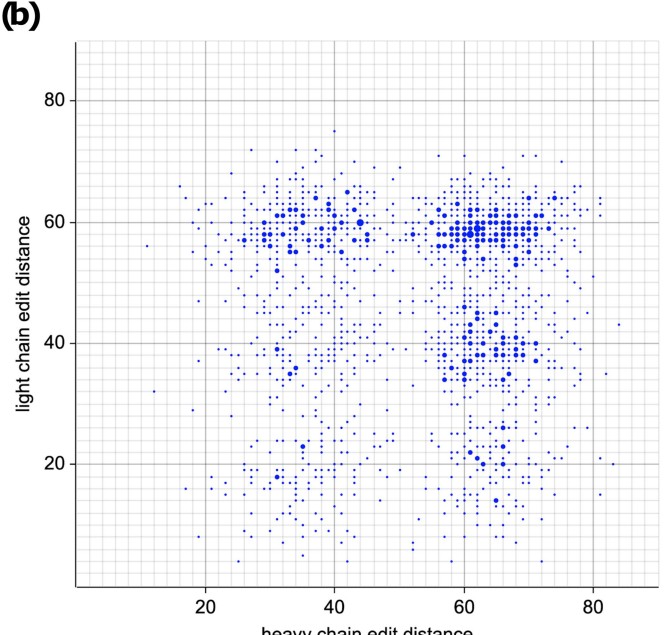

**Extended Data Fig. 2 | Light chain coherence is visible by sequence similarity.**
Each point represents a pair of memory cells from different donors. Heavy and light chain edit distances are plotted, using the amino acids starting at the end of the leader and continuing through the last amino acid in the J segment. Points with identical coordinates are combined by showing a large point whose area is proportional to the number of such points. **a**, Cell pairs are displayed if the two cells in the pair have the same CDRH3 amino acid sequence. To increase readability, only one third of such pairs were selected at random for display. Of the pairs, **78%** have light chain edit distance ≤ 20. This number (78%) is the fraction of cell pairs lying below the horizontal line at light chain edit distance 20, and was computed separately. It is proportional to the fraction of red below the line, if overlap is taken into account. **b**, [control] The same number of cell pairs were selected at random for display, without regard to CDRH3. Of the pairs, **9%** have light chain edit distance ≤ 20.

**Extended Data Table 1 | Donor information**

| Donor | Vendor | Catalog # | Relevant immune history (vendor donor ID) | Sex | Age | HLA type | Blood type | Known serologies |
|---|---|---|---|---|---|---|---|---|
| 1 | Cellero | 10052 | SARS-CoV-2 (523) | F | 45 | A*03:01/68:01; B*15:01/35:03; C*03:03/04:01; DRB1*13:01/16:01 | O+ | SARS-CoV-2 |
| 2 | Cellero | 1146 | SARS-CoV-2 (527) | F | 35 | A*02:01/02:01; B*44:02/51:01; C*02:02/05:01; DRB1*01:01/15:01 | O+ | SARS-CoV-2 |
| 3 | Cellero | 1132 | Type 1 Diabetes (607) | F | 38 | A*02:01/24:02; B*39:06/45:01; C*07:02/16:01; DRB1*04:04/11:01 | O+ | – |
| 4 | Cellero | 10050 | Celiac's disease (649) | F | 50 | A*02:01/30:02; B*18:01/51:01; C*02:02/05:01; DRB1*03:01/15:01 | O+ | CMV |

All donors were appropriately consented for genomic data use and release under protocols reviewed by independent IRB boards consulted by the vendor. Samples from the donors were tested by the vendor and confirmed to be both seronegative and not detectably infected with HIV-1, HIV-2, hepatitis B, hepatitis C, or HTLV-1. HLA typing, serology, and blood typing were also performed by the vendor. Donor 523 clinical timeline. Donor 523 tested positive for COVID-19 via RT-PCR of nasopharyngeal swabs on day 0 and was hospitalized from day −5 to day 0. She tested negative for COVID-19 at day 18, and had a plaque reduction neutralization test titer of 1:>2560 at day 44. She donated plasma and cells on day 65. Donor 527 clinical timeline. Donor 527 tested positive for COVID-19 via RT-PCR of nasopharyngeal swabs on day 0 and was not hospitalized. She tested negative for COVID-19 at day 15, and had a plaque reduction neutralization test titer of 1:20 at day 57. She donated plasma and cells on day 75.

**Extended Data Table 2 | Flow categories by donor with number of cells and naive fraction**

| Flow phenotypes | Cells | | | | | Percent naive | | | | |
|---|---|---|---|---|---|---|---|---|---|---|
| | all donors | donor 1 | donor 2 | donor 3 | donor 4 | all donors | donor 1 | donor 2 | donor 3 | donor 4 |
| Naive | 670,025 | 170,592 | 151,686 | 162,615 | 185,132 | 80.0 | 57.4 | 90.9 | 84.8 | 87.8 |
| Unswitched memory | 170,305 | 100,586 | 20,360 | 26,909 | 22,450 | 0.9 | 0.4 | 2.3 | 1.8 | 0.5 |
| Switched memory | 81,058 | 25,323 | 10,167 | 17,483 | 28,085 | 0.2 | 0.2 | 0.3 | 0.2 | 0.2 |
| Plasmablast | 9,443 | 2,502 | | 1,751 | 5,190 | 0.3 | 0.3 | | 0.9 | 0.1 |
| Memory subtotal | 260,806 | 128,411 | 30,527 | 46,143 | 55,725 | 0.6 | 0.3 | 1.7 | 1.2 | 0.3 |
| Unswitched + naive | 270,654 | | 53,929 | 72,966 | 80,559 | 56.5 | | 41.7 | 67.1 | 56.7 |
| Switched + naive | 207,454 | 65,700 | 57,390 | 71,025 | 76,539 | 54.1 | 19.2 | 75.4 | 69.5 | 53.9 |
| Total | 1,408,939 | 364,703 | 293,532 | 352,749 | 397,955 | 56.9 | 30.4 | 69.6 | 67.1 | 62.7 |

Total numbers of cells captured via fluorescence-activated flow cytometry with exactly two chains are shown, along with the fraction of naive sequences, as determined computationally by lack of exhibited SHM. Cells were sorted for naive, unswitched, switched and plasmablast, and in some libraries, sort categories were combined. Entries are blank if no data were generated. The table only accounts for cells that exhibited exactly one heavy and one light chain, and which were determined to lie in a valid clonotype having exactly two chains.

**Extended Data Table 3 | Light chain coherence does not imply heavy chain coherence**

| Memory B cells | | | | | | | | Naive B cells | | | | | | | |
|---|---|---|---|---|---|---|---|---|---|---|---|---|---|---|---|
| CDRH3 amino acid identity % | donors and percent heavy chain coherence | | | | | | | CDRH3 amino acid identity % | donors and percent heavy chain coherence | | | | | | |
| | any | 1,2 | 1,3 | 1,4 | 2,3 | 2,4 | 3,4 | | any | 1,2 | 1,3 | 1,4 | 2,3 | 2,4 | 3,4 |
| 0 | 5 | 4 | 2 | 3 | 8 | 17 | 5 | 0 | | | | | | | |
| 10 | 5 | 6 | 4 | 4 | 4 | 6 | 5 | 10 | | | | | | | |
| 20 | 5 | 5 | 4 | 5 | 4 | 5 | 4 | 20 | 5 | | 100 | | 0 | 0 | 0 |
| 30 | 5 | 5 | 5 | 5 | 5 | 5 | 4 | 30 | 3 | 4 | 0 | 1 | 4 | 4 | 2 |
| 40 | 5 | 5 | 5 | 5 | 5 | 5 | 4 | 40 | 3 | 4 | 2 | 2 | 2 | 4 | 2 |
| 50 | 5 | 5 | 5 | 5 | 5 | 5 | 5 | 50 | 4 | 4 | 4 | 5 | 4 | 5 | 4 |
| 60 | 5 | 5 | 5 | 5 | 5 | 5 | 5 | 60 | 4 | 4 | 4 | 4 | 4 | 4 | 4 |
| 70 | 5 | 5 | 5 | 5 | 5 | 6 | 5 | 70 | 4 | 4 | 4 | 4 | 4 | 4 | 4 |
| 80 | 5 | 5 | 5 | 6 | 6 | 6 | 5 | 80 | 4 | 4 | 4 | 4 | 4 | 4 | 4 |
| 90 | 6 | 5 | 5 | 6 | 6 | 7 | 6 | 90 | 4 | 4 | 4 | 5 | 4 | 4 | 4 |
| 100 | 6 | 6 | 6 | 7 | 7 | 7 | 6 | 100 | 5 | 5 | 5 | 5 | 4 | 4 | 4 |

Data are shown as in Fig. 1a, except that the role of heavy and light chains is reversed, and paralogs are not considered. Entries are blank in cases where there was no data because no heavy chain sequences could be compared at a given CDRH3 percent identity threshold. A value of 0 represents 0% heavy chain concordance.

**Extended Data Table 4 | Light chain coherence in memory B cells (public antibodies), without identifying light chain V gene paralogs**

| Memory B cells | | | | | | | | Naive B cells | | | | | | | |
|---|---|---|---|---|---|---|---|---|---|---|---|---|---|---|---|
| CDRH3 amino acid identity % | donors and percent light chain coherence | | | | | | | CDRH3 amino acid identity % | donors and percent light chain coherence | | | | | | |
| | any | 1,2 | 1,3 | 1,4 | 2,3 | 2,4 | 3,4 | | any | 1,2 | 1,3 | 1,4 | 2,3 | 2,4 | 3,4 |
| 0 | 5 | 4 | 5 | 5 | 4 | 3 | 5 | 0 | 4 | 4 | 4 | 4 | 4 | 3 | 4 |
| 10 | 5 | 5 | 5 | 5 | 4 | 4 | 5 | 10 | 4 | 4 | 4 | 4 | 4 | 5 | 4 |
| 20 | 5 | 5 | 5 | 5 | 5 | 5 | 5 | 20 | 5 | 5 | 5 | 5 | 5 | 5 | 5 |
| 30 | 5 | 5 | 6 | 5 | 5 | 5 | 5 | 30 | 5 | 5 | 5 | 5 | 5 | 5 | 5 |
| 40 | 6 | 6 | 6 | 6 | 5 | 5 | 5 | 40 | 5 | 5 | 5 | 5 | 5 | 5 | 5 |
| 50 | 7 | 7 | 7 | 7 | 6 | 6 | 6 | 50 | 5 | 5 | 6 | 5 | 5 | 5 | 5 |
| 60 | 12 | 14 | 14 | 12 | 11 | 10 | 11 | 60 | 6 | 6 | 6 | 6 | 6 | 6 | 6 |
| 70 | 29 | 37 | 29 | 28 | 26 | 24 | 25 | 70 | 6 | 6 | 6 | 6 | 6 | 6 | 6 |
| 80 | 58 | 61 | 60 | 60 | 53 | 51 | 54 | 80 | 7 | 7 | 7 | 7 | 6 | 6 | 7 |
| 90 | 62 | 65 | 71 | 51 | 75 | 62 | 68 | 90 | 8 | 7 | 9 | 8 | 7 | 8 | 8 |
| 100 | 64 | 68 | 53 | 60 | 86 | 79 | 65 | 100 | 9 | 9 | 7 | 12 | 7 | 10 | 9 |

Data are shown as in Fig. 1a, except that light chain V gene paralogs are not treated as the same.

# Reporting Summary

## Statistics

For all statistical analyses, confirm that the following items are present in the figure legend, table legend, main text, or Methods section.

| n/a | Confirmed | |
|---|---|---|
| ☐ | ☒ | The exact sample size (*n*) for each experimental group/condition, given as a discrete number and unit of measurement |
| ☐ | ☒ | A statement on whether measurements were taken from distinct samples or whether the same sample was measured repeatedly |
| ☐ | ☒ | The statistical test(s) used AND whether they are one- or two-sided *Only common tests should be described solely by name; describe more complex techniques in the Methods section.* |
| ☒ | ☐ | A description of all covariates tested |
| ☒ | ☐ | A description of any assumptions or corrections, such as tests of normality and adjustment for multiple comparisons |
| ☐ | ☒ | A full description of the statistical parameters including central tendency (e.g. means) or other basic estimates (e.g. regression coefficient) AND variation (e.g. standard deviation) or associated estimates of uncertainty (e.g. confidence intervals) |
| ☐ | ☒ | For null hypothesis testing, the test statistic (e.g. *F*, *t*, *r*) with confidence intervals, effect sizes, degrees of freedom and *P* value noted *Give P values as exact values whenever suitable.* |
| ☒ | ☐ | For Bayesian analysis, information on the choice of priors and Markov chain Monte Carlo settings |
| ☒ | ☐ | For hierarchical and complex designs, identification of the appropriate level for tests and full reporting of outcomes |
| ☒ | ☐ | Estimates of effect sizes (e.g. Cohen's *d*, Pearson's *r*), indicating how they were calculated |

*Our web collection on statistics for biologists contains articles on many of the points above.*

## Software and code

Policy information about availability of computer code

| Data collection | Sony MA900 software used to acquire flow cytometry data. |
|---|---|
| Data analysis | All code to replicate key findings and process data in this study are available at https://github.com/DavidBJaffe/enclone (Git hash 561e3ac). An additional copy of all code for analysis and visualization, including source data, can be found at DOI 10.25452/figshare.plus.20338177. We used version 0.5.175 of enclone, version 1.29.2 of the msa R package, and version 0.1 of the ggseqlogo R package. |

For manuscripts utilizing custom algorithms or software that are central to the research but not yet described in published literature, software must be made available to editors and reviewers. We strongly encourage code deposition in a community repository (e.g. GitHub). See the Nature Portfolio guidelines for submitting code & software for further information.

## Data

Policy information about availability of data

All manuscripts must include a data availability statement. This statement should provide the following information, where applicable:
- Accession codes, unique identifiers, or web links for publicly available datasets
- A description of any restrictions on data availability
- For clinical datasets or third party data, please ensure that the statement adheres to our policy

All data are publicly available and processed at DOI 10.25452/figshare.plus.20338177. See Methods for informed consent and related information.

# Human research participants

Policy information about studies involving human research participants and Sex and Gender in Research.

| | |
|---|---|
| Reporting on sex and gender | Samples from biological male and female participants were used as part of this work. Additional information is provided in the Extended and Supplementary Information. |
| Population characteristics | This information is provided in Extended Tables. |
| Recruitment | Participants were recruited independently by third party vendors, throughout the United States. It is unlikely that particular selection biases were present given the genetic diversity of the individuals in this study, and representation of individuals from multiple continents. |
| Ethics oversight | An independent IRB providing oversight to the vendor approved the collection protocol and informed consent for all samples for all donors. |

Note that full information on the approval of the study protocol must also be provided in the manuscript.

# Field-specific reporting

Please select the one below that is the best fit for your research. If you are not sure, read the appropriate sections before making your selection.

☒ Life sciences ☐ Behavioural & social sciences ☐ Ecological, evolutionary & environmental sciences

For a reference copy of the document with all sections, see nature.com/documents/nr-reporting-summary-flat.pdf

# Life sciences study design

All studies must disclose on these points even when the disclosure is negative.

| | |
|---|---|
| Sample size | We chose samples from genetically diverse individuals who were unlikely to be related to each other. As this was an exploratory study and traditional power calculations are not applicable to immune repertoire data, we did not perform or invent a power calculation. |
| Data exclusions | No data were excluded from the study. |
| Replication | All central results related to light chain coherence were successfully replicated within and between an additional 25 donors, including an additional 2 donors whose data were independently generated by an group as part of an independent prior study (Phad et al. 2022 Nat. Immunol.). Biological replicates (separate cell aliquots) were used from each of the four main donors. |
| Randomization | Donors were not randomized to experimental groups as part of this work; this work is not a clinical study and does not describe an intervention-based study. We used samples from 4 separate and genetically unrelated individuals, in additional to using previously published independent data, to assess our results. |
| Blinding | We did not blind ourselves as part of this study because no randomization was performed. |

# Reporting for specific materials, systems and methods

We require information from authors about some types of materials, experimental systems and methods used in many studies. Here, indicate whether each material, system or method listed is relevant to your study. If you are not sure if a list item applies to your research, read the appropriate section before selecting a response.

## Materials & experimental systems

| n/a | Involved in the study |
|---|---|
| ☐ | ☒ Antibodies |
| ☒ | ☐ Eukaryotic cell lines |
| ☒ | ☐ Palaeontology and archaeology |
| ☒ | ☐ Animals and other organisms |
| ☒ | ☐ Clinical data |
| ☒ | ☐ Dual use research of concern |

## Methods

| n/a | Involved in the study |
|---|---|
| ☒ | ☐ ChIP-seq |
| ☐ | ☒ Flow cytometry |
| ☒ | ☐ MRI-based neuroimaging |

# Antibodies

| | |
|---|---|
| Antibodies used | Described in Methods in subsection "Flow cytometry." Clone, color, isotype, catalog #, host, and manufacturer are provided. All antibodies were used at 1:40 dilution (5 ul per antibody) in a staining volume of 200 ul with a total antibody mass of 1.25 ug. |
| Validation | All antibodies used in this study are GMP manufactured and have been verified by 3 independent manufacturers and vendors using relative expression (specific staining of known lineage marker-positive cells, no staining of lineage marker-negative cells) with imaging and flow cytometry. All antibodies used in this study have been used extensively in the literature for flow cytometric analysis, including in specialist journals such as Cytometry Part A. |

# Flow Cytometry

## Plots

Confirm that:

☒ The axis labels state the marker and fluorochrome used (e.g. CD4-FITC).

☒ The axis scales are clearly visible. Include numbers along axes only for bottom left plot of group (a 'group' is an analysis of identical markers).

☒ All plots are contour plots with outliers or pseudocolor plots.

☒ A numerical value for number of cells or percentage (with statistics) is provided.

## Methodology

| | |
|---|---|
| Sample preparation | Described in detail in Methods. Briefly, PBMCs from each donor were thawed, washed, stained, analyzed, and sorted using standard protocols and validated antibody panels. See Phad et al. 2022 for methods information related to publicly available data first reported outside of this study. |
| Instrument | Sony MA-900 |
| Software | Sony MA-900 software; FlowJo used for visualization of data. |
| Cell population abundance | Reported within figures. |
| Gating strategy | Reported both in Methods and in Figures. |

☒ Tick this box to confirm that a figure exemplifying the gating strategy is provided in the Supplementary Information.

