## [Peer Review File · Nature]

Manuscript Title: Functional antibodies exhibit light chain coherence

Reviewer Comments & Author Rebuttals

Reviewer Reports on the Initial Version:

Referees' comments:

Referee #1 (Remarks to the Author):

A. Summary of the key results

Jaffe et al. analyzed the distribution of paired heavy and light chain pairs in functional B cell receptors from four human donors. The primary claim resulting from this investigation was that unrelated B cells with similar heavy chain rearrangements tended to have similar light chain rearrangements, in a phenomenon they refer to as "light chain coherence." They provide evidence that this coherence is a feature of memory B cells, but not naïve B cells, occurring both within and between donors.

The extent of light chain coherence was reported to be very high, with >80% of cell pairs from separate donors with 100% CDRH3 amino acid identity having coherent light chains. If this property is indeed a broad and general feature of B cell repertoires, then there is some novelty and utility to the observation. However, it is perhaps not that surprising. Both the heavy and light chain are involved in determining B cell specificity to many antigens. The observed light chain coherence could simply reflect that the four individuals studied were exposed to a similar pathogenic environment (we are told the donors are unrelated, but not much else). In addition, there are several major concerns with the methods that detract from the significance of the findings. Finally, several additional analyses were included in the manuscript whose relationship to the central claim of light chain coherence was unclear.

B. Originality and significance: if not novel, please include reference

Major concerns about the methods employed:

1. Clonal structure. B cells belong to "clones", which arise from a common VDJ rearrangement. If sequences from a clone are treated independently this could overestimate the extent of light chain coherence. The authors demonstrate light chain coherence across subjects; which they claim bypasses the issue of clonal structure. However, because sequences are still treated independently within each subject, it is possible their results are explained by a small number of expanded clones within the subjects. This possibility is suggested by some of the numbers. There are 2,813 memory cells with 100% CDR3 aa identity (line 97), but only ~5,000 cell pairs across subjects with 100% CDR3 aa identity. This suggests that a small number of donors have a higher proportion of these cells that match with small numbers of cells in other donors. These could be clonal expansions within each subject. Meaning that the results shown could be driven by only a handful of VDJ recombination

events. To address this, the authors should identify clonal families within each subject, and ensure each clone is represented by a single B cell. The number of cells and cell pairs from each donor in Fig. 1 should also be reported. Further, for Fig 2, the authors only compare cells with different V genes and claim this addresses the issue of clonal expansions. However, it is possible that two cells from the same clone have accumulated enough somatic hypermutations that their heavy chain V genes are assigned different (or ambiguous) V gene names. This possibility should be refuted, perhaps by showing the V genes of these pairs are far more different than expected from somatic hypermutation alone. Finally, the authors should report the number of cells from each patient and cell pairs used in each comparison, not just the total number of pairs, for Figs. 1 and 2. Fig. 2 currently does not even report the number of cell pairs.

2. Contamination and small sample size. The central claim is that light chain coherence is a general feature of B cell biology; however, it is only demonstrated in four subjects processed by a single lab. It is not clear what steps the authors take to guard against contamination between samples. The authors should show that (1) the IGHV/J nucleotide sequences of the coherent cell pairs are not identical, and (2) that the distance between the cell barcodes of coherent cell pairs follows a similar distribution to non-coherent pairs. In addition, the author's claims would be significantly strengthened by demonstrating that light chain coherence also occurs between their four subjects and other subjects from previously published datasets from independent labs. If their results are truly general, there should be significant light chain coherence.

3. B cell sorting. Multiple B cell subsets are sorted from each sample. However, only the results for total memory and naive B cells are detailed. The authors should divide their results by their original sorting strategy to make it clear they are consistent between these subgroups.

Other studies have broached the area of light chain coherence and are not cited or discussed. For example:

- Biases in heavy/light chain pairing: <https://academic.oup.com/peds/article/25/10/523/1559655>
- Human B cell clones tend to use the same light chain rearrangement: <https://pubmed.ncbi.nlm.nih.gov/31484734/>

Finally, beyond the central claims of light chain coherence made in Figs 1 and 2, the authors include multiple additional analyses that are of unclear relevance. Specific critiques of these additional analyses are detailed in comments below.

C. Data & methodology: validity of approach, quality of data, quality of presentation

1. Line 76: For defining naïve B cells, why is only the level of somatic hypermutation used and not also the constant region? Naïve B cells should not have switched isotypes. Also, it is unclear why the term (dref) is used instead of a more interpretable term for somatic hypermutation frequency. It is also unclear how dref differs from "substitution rate" later on.
2. Lines 92-94. How much of a difference does grouping these V genes make? Some reasoning as to why they were considered these V gene paralogs to be identical should be shown.
3. Line 126. Are ambiguous V gene assignments counted as similar or different? If ambiguous V

genes count as different, this could unintentionally include sequences from the same clonotype.

4. Line 135-142. It is not clear why or how the substitution matrix analysis was performed. What is the substitution matrix intended to be used for? If it is intended to improve clonal clustering or other measures, no convincing attempt is made to show this. If it is to investigate the properties of amino acid substitutions, no attempt is made at that either. The methods for this section leave many important details unspecified. For instance, what sequences are being compared for computing the distance?

a. In addition to this, when perturbing the matrix, what is the range of values that the randomly chosen entry is increased/decreased by?

b. Why is the max value of the matrix capped at 8?

5. Line 150-152. It seems as though the authors fit the substitution model to their data and then show that the model has an improved fit to the same data. It is unclear what is to be concluded from this result.

6. Line 159. Typo: "and if Y is similar to Z, they both go in the same group, and therefore X and Z must go in the same group..."

7. Line 163-169. Are these initial groupings defined using only the heavy chain? This should be specified. The results of this analysis are inadequately detailed in the text. For instance, what is the light chain coherence for transitive clonotypes? We are only told that it is lower than in Figs. 1-2.

8. Lines 214-242. In this section, the authors appear to be testing their hypothesis that recurrent clones occur across donors purely by chance. They test this by performing simulations with OLGA. Their simulations showed significantly fewer recurrent clones than their empirical data, in opposition to their hypothesis. Rather than changing their conclusions, the following paragraph appears to imply that their hypothesis was correct but that the model used in simulations was improperly specified.

9. Line 231. What is the substitution rate, and how does it differ from somatic hypermutation level? If naive sequences are being simulated, why are there any substitutions at all?

10. Line 236. Is 17.8% the true value of simulated VDDJ junctions, or is that the value estimated from the analysis pipeline? It would need to be the latter to be directly comparable to the real naive antibodies. How are VDDJ junctions inferred and how accurate is that process?

11. Lines 244-250. It is unclear what analysis was done here, what the results were, and what they mean in the context of the other analyses.

12. Fig 1: Can the authors speculate as to why the number of memory and naive cell pairings is roughly the same for each bin?

D. Appropriate use of statistics and treatment of uncertainties

There does not appear to be any formal treatment of significance or uncertainty in the results presented. For example, the central claim of ~80% light chain coherence in memory B cells is presented as being surprising. However, the null hypothesis for why this result is noteworthy is never explicitly stated or calculated. If heavy and light chains are paired together randomly during initial V(D)J recombination, what level of coherence would be expected? At the very least, the results should be tested through a permutation test. Naive B cells are not an appropriate null hypothesis because the V gene usage of the memory B cell pool is expected to be far more biased (for example, <https://doi.org/10.1182/blood-2010-03-275859>). The authors should add a control line to Fig.1 that include randomly chosen cells with the same V/J gene and CDRH3 length distribution to

match each of the points on the red lines. Additional biases are likely to be induced by the clonal relationships within each of the donors, as mentioned above.

E. Conclusions: robustness, validity, reliability

Our major critiques of the conclusions are given in the “significance and originality section.”

F. Suggested improvements: experiments, data for possible revision

1. Remove all italics used for emphasis. All terms and analyses should be adequately specified to be understandable without italics.
2. Extended Figure 4 is a nice data visualization, but it is difficult to see how the percentages presented in the text relate to the figure.
3. In Figures 3a and 3b, remove the ‘a’ symbol from the ggplot legend.
4. The referencing of the figures is inconsistent. It seems to shift from, for example, Figure 3(a) to Figure 4a depending on the figure.
5. It is known that both the heavy and light chain are involved in determining B cell specificity to many antigens. The observed light chain coherence could simply reflect that the four individuals studied were exposed to related pathogens. Can the authors provide information on whether these individuals were chosen from similar populations? Would the same light chain coherence be expected from individuals sampled from dramatically different environments?

G. References: appropriate credit to previous work?

1. Lines 30-38: there are multiple claims made in this paragraph, none of which have citations to support them.
2. Lines 121-123: while this is true, please provide a source to support your claim.
3. Lines 131-132 why do you expect the true rate of coherence to be approximately equal?
4. Lines 136-137 why does grouping that treats all amino acids as equal lead to nonoptimal results?
5. Line 236 (held within parentheses) please provide a reference that a VDDJ antibody would be far less likely to recur by chance.
6. Lines 256-257: please provide a source to support this claim
7. Lines 308-309: please provide a source to support this claim.
8. The authors introduce “transitive” clonal grouping, which does not appear any different from single linkage hierarchical clustering, which is commonly practiced in the field to identify clonal groups.

H. Clarity and context: lucidity of abstract/summary, appropriateness of abstract, introduction and conclusions

The abstract appropriately summarizes the key findings from Figs. 1 and 2. However, the claim that “naive antibodies recur by chance” does not appear to be supported within the text, and in fact appears to be undermined by the simulation analyses in Lines 214-224 (see comments above). The results section concludes with the sentence: “Indeed, our findings imply that all antibodies are recurrent, but at varying rates depending on their junction complexity and the prevalence of their

cognate antigen.” This is not a meaningful statement, considering that rates of recurrence could vary between 0 and 100% percent.

Further, in the statement, “Our work reveals that the light chain of functional antibodies is highly constrained”, the authors appear to use the term “functional” to mean “not naive.” This is confusing, since within B cell repertoire analysis, “functional” is typically used to mean BCRs that encode a functional protein. This should be clarified throughout.

Referee #2 (Remarks to the Author):

In this excellent study, the authors performed an extremely deep sequencing of the circulating human B cell repertoire, producing a dataset of ~1.6M natively paired BCR sequences. Large sequencing sets have been collected previously but not retaining native pairing. In addition to generating (and already making publicly available!!), an unprecedented dataset, the authors report the existence of light chain coherence in public, antigen-experienced BCRs. This is a fundamental immunological insight that could only be observed by examining a dataset of this size.

The manuscript is clear and well written, although a few areas could be a bit more concise. Also, the clarity of the data could be improved by converting a few tables to figures. For example, plotting the data in Figure 4e would make the differences between naive and simulated repertoires much easier to distinguish. Of course, the authors could retain the tables as supplementary material so that the quantitative data is still accessible if desired.

Minor issues:

- The authors use a novel method for allele inference, however there is some confusion about the methodology. First, after performing a V-gene pileup, the authors identify positions of potential allelic variance (>4 occurrences and $\geq 25\%$ of the total pileup) and then mention that “each cell has a footprint relative to these positions”. It is not clear what the authors mean by footprint — are residues surrounding the potentially variant position somehow also being considered variants if they are also non-reference? Second, did the authors benchmark their allelic inference method against other inference methods? If not, such a comparison would seem appropriate— not to make any claims about the accuracy of any of the methods, but merely to give readers (who may be more familiar/comfortable with existing tools) a sense of reference. This is particularly relevant because the computational splitting of cells into naive and memory subsets uses a threshold of zero mutations for naive B cells, which would be highly sensitive to differences in allele inference.

For the coherence optimized substitution matrix, is there a reason the authors selected a range of [0, 8] for the dissimilarity values rather than re-scaling to [0, 1]?

In Figure 4d (and the associated text) the authors mention “junction substitution rate” when discussing junctions with no N-addition at VD or DJ junctions. It is not clear what the authors mean by substitution rate, as these antibodies are all naive and would not be expected to contain SHM-induced substitutions.

When performing sequence simulation with OLGA, it appears the authors used the default simulation settings, which were modeled on non-productive recombinations to isolate the specific likelihood of VDJ recombination events and minimize skewing caused by negative selection during early B cell development. As such, it is not necessarily surprising that the synthetic repertoires are

somewhat distinct from observed naive repertoires. It may be useful to clarify this, which would strengthen the authors conclusion — if negative selection does indeed “sieve” the repertoire by removing unacceptable recombinations, it is likely that the sieved repertoires would be more similar across individuals than would be expected from coincident recombinations alone.

Referee #3 (Remarks to the Author):

Jaffe and colleagues described an analysis of single cell sequencing data of human antibody repertoires and addressed a long-standing question about specifics of heavy and light chain pairings. The textbook description of antibody repertoire generation tells that heavy and light chain pairing increases the diversity of antibodies by several orders of magnitude. The authors update this statement and show that while the naive diversity of heavy and light chain pairs is high, it is significantly reduced in mature B cells. This finding is extremely important for understanding features of functional antibodies and has applications in vaccine and drug design.

I read the manuscript with great interest. It is very well written, the conclusions are clear and justified. The cited papers are appropriate.

I have several clarifying questions about the performed methods and experiments. I agree with the general layout, but I would be grateful for additional details - please see questions 1–4. I also think that additional computational experiments described in questions 4–7 will be interesting for immunologists.

Major comments:

1. While it's known that flow cytometry can create both false positives and false negatives, the computational method described in lines 72–84 also can introduce artificial biases. E.g., V(D)J sequences that were classified as naive by flow cytometry can contain mutations wrt germline genes because they were derived from unknown alleles of germline genes. Similarly, V(D)J sequences that were classified as memory can be mutation-free because corresponding germline genes do not need mutations to recognize certain types of antigens. E.g., anti-SARS-CoV-2 antibodies reported by Kim et al., 2021 (<https://doi.org/10.1126/scitranslmed.abd6990>) have these features. While the frequency of this phenomena has not been studied, it's important to take it into account. I would recommend adding a remark on possible computational biases and comment on their impact on the obtained results.

2. Lines 60–62 “We exclude related cells (i.e., those in the same clonotype) because they use the same VDJ genes by definition” and following them lines 88–90: “We only considered pairs of cells with the same heavy chain V gene and the same CDRH3 length, and whose cells came from different donors”.

Could authors clarify how clonal expansion has been taken into account? Does the sentence in lines 60–62 mean that authors keep only one cell from each clonotype? Let's assume that Donor 1 and Donor 2 have clonal lineages with similar heavy chains formed M and N cells, respectively. Does it mean that authors analyze MxN pairs of cells or only one pair? If MxN pairs are considered, then they can introduce artificial biases caused by overrepresentation of pairs of cells with similar chains.

3. A follow-up question: Let's again assume that Donor 1 and Donor 2 have clonal lineages with similar heavy chains formed M and N cells, respectively. What is the coherence inside clonal lineages? Is it 100%? Do authors observe dual light chains (doi:10.1016/j.coi.2014.01.012)?

4. Lines 90–91: “We divided the pairs into eleven sets by their CDRH3 amino acid percent identity, rounded down to the nearest 10%”. Could authors describe how large these clusters are and how many donors each cluster represents?

5. Could authors add an analysis of heavy and light V gene pairings? I think a heatmap showing IGHV genes as rows and light chain V genes as columns (a heatmap cell = the fraction of clonotypes with a pair of V genes) would be illustrative. Which IGHV have strong preferences toward light chain V genes and which are promiscuous? Such analysis can be useful for antibody drug design.

6. A similar analysis of CDRH3 lengths and light chain V genes could be also very useful.

7. Do existing antibody structures confirm findings described by authors?

Minor:

Lines 51–52: “four heavy chain V genes and two light chain V genes” - please list those genes.

Referee 1 comments

“However, it is perhaps not that surprising. Both the heavy and light chain are involved in determining B cell specificity to many antigens. The observed light chain coherence could simply reflect that the four individuals studied were exposed to a similar pathogenic environment (we are told the donors are unrelated, but not much else).”

We agree that in present times, it would be typical for two arbitrary adults to have experienced during their lives shared exposure to a panoply of diverse antigens. We certainly agree that our observations reflect such exposure! However it does not follow that the acquired antibodies are the same. Certainly many heavy chains are observed for the same antigen. Thus light chain coherence is not an automatic consequence of common exposure. It just happens that that is how nature works, as we demonstrate.

In the original submission, we also provided HLA types and documented serologies for the four individuals, which are indicative of diverse ancestry and some common immune exposures. Still we understand the concern that somehow these individuals are special, and as requested we describe results for other individuals (as separately requested by referee #1, later in this response). Separately, the donors from Phad et al. 2022 *Nat. Immunol.* are Swiss and biologically male, 50 and 69 years old respectively, and received separate influenza vaccinations in 2009 and 2010, per publication.

“Clonal structure. B cells belong to “clones”, which arise from a common VDJ rearrangement. If sequences from a clone are treated independently this could overestimate the extent of light chain coherence. The authors demonstrate light chain coherence across subjects; which they claim bypasses the issue of clonal structure. However, because sequences are still treated independently within each subject, it is possible their results are explained by a small number of expanded clones within the subjects. This possibility is suggested by some of the numbers. There are 2,813 memory cells with 100% CDR3 aa identity (line 97), but only ~5,000 cell pairs across subjects with 100% CDR3 aa identity. This suggests that a small number of donors have a higher proportion of these cells that match with small numbers of cells in other donors. These could be clonal expansions within each subject. Meaning that the results shown could be driven by only a handful of VDJ recombination events. To address this, the authors should identify clonal families within each subject, and ensure each clone is represented by a single B cell.”

We understand and agree with this logic, and yet at the same time power is lost by reducing to just one cell per clonotype. For example, frequently cells from different donors that have 100% amino acid identity on CDRH3 are lost. Therefore we do the analysis both ways (using all cells in a clonotype, and using just one cell; a number of other ways are also shown in **Supplementary Table 2**) We show below the new **Figure 1** data if one cell is used, noting that the change from the original figure is small.

Memory B cells; 1 cell per clonotype								
CDRH3 amino acid % identity	log ₁₀ of cell pairs	Donors and percent light chain coherence						
		any	1,2	1,3	1,4	2,3	2,4	3,4
0%	5.2	5	5	5	5	5	4	5
10%	7.0	5	5	5	5	4	5	5
20%	7.8	5	5	5	5	5	5	5
30%	8.0	5	5	5	5	5	5	5
40%	7.7	5	6	6	5	5	5	5
50%	7.3	6	6	7	6	6	6	6
60%	6.4	8	9	10	8	9	7	8
70%	5.5	17	19	18	16	18	15	15
80%	4.6	38	42	37	41	38	35	33
90%	3.7	63	62	61	65	63	64	58
100%	3.0	79	74	74	84	80	83	75

Naive B cells; 1 cell per clonotype								
CDRH3 amino acid % identity	log ₁₀ of cell pairs	Donors and percent light chain coherence						
		any	1,2	1,3	1,4	2,3	2,4	3,4
0%	4.5	4	4	4	4	4	4	4
10%	7.0	5	5	5	5	4	5	4
20%	8.2	5	5	5	5	5	5	5
30%	8.4	5	5	5	5	5	5	5
40%	8.2	5	5	5	5	5	5	5
50%	7.8	5	5	6	6	5	5	5
60%	7.1	6	6	6	6	6	6	6
70%	6.3	6	6	6	6	6	6	6
80%	5.3	7	7	7	7	7	6	7
90%	3.9	8	7	9	9	7	8	8
100%	2.6	10	9	7	10	9	10	10

“Further, for Fig 2, the authors only compare cells with different V genes and claim this addresses the issue of clonal expansions. However, it is possible that two cells from the same clone have accumulated enough somatic hypermutations that their heavy chain V genes are assigned different (or ambiguous) V gene names. This possibility should be refuted, perhaps by showing the V genes of these pairs are far more different than expected from somatic hypermutation alone.”

The referee is totally right. Repeated events must be present within single donor datasets, but finesse is required to extract them without being tricked by under-clonotyping, and the referee was correct that sometimes reference sequences were misassigned. We therefore revised our analysis to greatly mitigate this possibility.

In the revised analysis, instead of assuming that heavy chain genes are different, we assumed that they are the same, and required a separate validation condition for a cell pair, to ensure (with fairly high likelihood) that the cells in the pair arise from different clonotypes. At least one of three conditions had to be satisfied:

1. The data support different heavy chain J gene usage for the two cells. For this, we examined framework region four for both cells. There had to exist at least **three** positions where the reference sequences were different, and the cells had bases consistent with those, and **no** positions where the reference sequences were different, and the cells both supported just one of the references.
2. Same thing but for the light chain instead of the heavy chain.
3. The light chain CDR3 lengths differed.

It is true that any of these conditions could happen by chance *within* a clonotype, but the odds of this significantly affecting the results are slim.

With the new analysis, and using only one cell per clonotype, here are the revised data for Figure 2:

Memory B cells							Naive B cells						
CDRH3 amino acid % identity	log ₁₀ of cell pairs	Donors and percent light chain coherence					CDRH3 amino acid % identity	log ₁₀ of cell pairs	Donors and percent light chain coherence				
		All	1	2	3	4			All	1	2	3	4
0%	4.8	4	5	4	5	3	0%	4.1	4	4	3	4	4
10%	6.5	5	5	4	4	4	10%	6.6	5	5	5	5	5
20%	7.4	5	5	4	5	5	20%	7.8	5	5	5	5	5
30%	7.6	5	5	5	5	5	30%	8.0	5	5	5	5	5
40%	7.3	5	5	5	5	5	40%	7.8	5	5	5	5	5
50%	6.9	6	6	5	5	5	50%	7.5	5	5	5	5	5
60%	6.1	8	9	7	7	6	60%	6.8	5	6	5	6	5
70%	5.2	15	18	14	12	12	70%	6.0	6	6	6	6	6
80%	4.3	36	42	33	25	32	80%	5.0	6	8	6	6	6
90%	3.5	60	61	55	55	62	90%	3.6	8	11	7	8	8
100%	3.0	65	66	70	66	62	100%	2.5	12	16	12	12	10

“Finally, the authors should report the number of cells from each patient and cell pairs used in each comparison, not just the total number of pairs, for Figs. 1 and 2. Fig. 2 currently does not even report the number of cell pairs.”

Yes these data should be included, and they are now part of **Supplementary Table 2**.

“The author’s claims would be significantly strengthened by demonstrating that light chain coherence also occurs between their four subjects and other subjects from previously published datasets from independent labs. If their results are truly general, there should be significant light chain coherence.”

This makes total sense:

1. We compiled a heterogeneous collection of older single cell VDJ datasets from diverse sources (from many individuals, Methods addition) and used those in place of the datasets of this work. We merged datasets in cases where index swapping might have introduced artifacts (per reviewer one’s point about contamination), as in fact in these data there **was** contamination! A detailed summary of the data is provided as **Supplementary Table 1**. They comprise **280,699** cells from **23** donors (or super-donors, because of merging). For these data, at 100% CDRH3 amino acid identity, we find **93%** light chain coherence (or 87%, if one cell per clonotype is used). The particularly high light chain coherence might be explained by selection for binding to particular antigens in some of the data.
2. Very recently, a new dataset appeared in Phad et al. 2022 *Nat. Immunol.* These data comprise **247,516** cells from **2** donors (approximately evenly distributed). For these data, at 100% CDRH3 amino acid identity, we find **79%** light chain coherence (or 69%, if one cell per clonotype is used).

Further, we conducted an analysis based on three “super-donors”. In this analysis, all of the older data was treated as one donor, and all of the Phad data was treated as one donor, and all of our data was treated as one donor. With these assumptions, at 100% CDRH3 amino acid identity, we find **70%** light chain coherence (or 77%, if one cell per clonotype is used). Much more detail is provided as a supplemental document.

These analyses support our claim that there is “nothing special” about the individuals chosen for this work, and that light chain coherence is a generalizing phenomenon in human antibody repertoires.

“The central claim is that light chain coherence is a general feature of B cell biology; however, it is only demonstrated in four subjects processed by a single lab. It is not clear what steps the authors take to guard against contamination between samples. The authors should show that (1) the IGHV/J nucleotide sequences of the coherent cell pairs are not identical, and (2) that the distance between the cell barcodes of coherent cell pairs follows a similar distribution to non-coherent pairs.”

This is a good question, as we have seen other data that are contaminated. We have several lines of evidence to demonstrate that contamination is not driving the results of this work:

1. In our experience, contamination is associated most commonly with older data that predate dual indexing.
2. When contamination did occur in older large datasets, we would find clonotypes containing cells from different donors, and having identical sequences. In the current data, the closest two cells from different donors get is 10 (nucleotide distance).
3. We also observe light chain coherence in the data from Phad et al. 2022, and for those data, the distance as above is 18.
4. The naive cells are a control. If contamination were driving results for memory cells, we would expect the same thing to happen for naive cells, but it does not.
5. See super-donor analysis in previous response.

The fifth point establishes that light chain coherence occurs even across datasets generated at separate institutions, where contamination is impossible.

B cell sorting. Multiple B cell subsets are sorted from each sample. However, only the results for total memory and naive B cells are detailed. The authors should divide their results by their original sorting strategy to make it clear they are consistent between these subgroups.

This is a great question. We added the numbers for switched memory and unswitched memory. They are about the same. We did not do the analysis for plasmablast cells; although they are even more interesting, there are not enough cells in our data to support this analysis.

Other studies have broached the area of light chain coherence and are not cited or discussed. For example:

- Biases in heavy/light chain pairing:

<https://academic.oup.com/peds/article/25/10/523/1559655>

We thank the referee for referring us to this paper, which examines a total of 545 antibodies that were present in the PDB circa 2012. Jayaram *et al.* used chi-squared tests with Bonferroni corrections to identify putatively significant associations between heavy chain V genes and light chain V genes. They found an association of IGHV1 with IGKV3 (which we replicate in Supplementary Figure 1). **However, the authors concluded that "pairing preferences do exist in the germline, but only for a small proportion of germline gene segments."** We believe our results directly contradict this statement by instead framing the problem through the view of 1.4 million paired junction regions from single cells, which is unsurprising given the sample size available to Jayaram *et al.* over a decade ago (n=545 human antibody sequences). The results described by Jayaram *et al.* are not directly comparable or relevant to our work, as they attempted to back-translate the protein sequences of 545 human antibodies (no antibody genetic information is recorded as part of PDB data

deposition), and because their work does not in fact approach or describe light chain coherence through the view of the heavy chain junction.

- Human B cell clones tend to use the same light chain rearrangement:
<https://pubmed.ncbi.nlm.nih.gov/31484734/>

This is an important paper for understanding clonotyping, but does not appear to broach the topic of light chain coherence. It describes a method for grouping cells into clonotypes, which ought to by definition have coherent light chains as the authors are attempting to identify clonally related cells. No comparisons within or between donors were made by the authors.

Line 76: For defining naïve B cells, why is only the level of somatic hypermutation used and not also the constant region? Naïve B cells should not have switched isotypes.

The number of class switched cells *without* SHM in our data is small (here showing only the pure libraries):

	donor				
flow sort class	all	d1	d2	d3	d4
naive	9	4	0	1	4
unswitched	92	49	11	15	17
switched	138	47	23	31	37
plasmablast	12	2		6	4

In total, these cells represent a mere 0.05% (n=749 single cells) of our entire dataset (here showing cells with no SHM and non-IgM/IgD isotypes in the dataset; see below/next page):

	donor				
isotype	all	d1	d2	d3	d4
Unknown	279	19	138	32	90
IgG1	100	23	49	10	18
IgG2	16	4	5	5	2
IgG3	26	5	11	3	7
IgA1	147	45	18	39	45
IgA2	181	111	5	15	50
All	749	207	226	104	212

We modified one sentence (shown as underlined):

*“We labeled an antibody sequence as **naïve** if it had no mutations relative to the inferred germline (i.e. exhibited no SHM), and as **memory** otherwise, understanding that these categories are biological oversimplifications and do not account e.g. for class switching.”*

We thought it better to be up front about the limitations, and given the small number of cells in this category, and that we don't understand them particularly well, it did not seem helpful to go further.

Also, it is unclear why the term (dref) is used instead of a more interpretable term for somatic hypermutation frequency. It is also unclear how dref differs from "substitution rate" later on.

We changed " $d_{ref} = 0$ " to "exhibited no SHM" and " $d_{ref} > 0$ " to "exhibited SHM".

Lines 92-94. How much of a difference does grouping these V genes make? Some reasoning as to why they were considered these V gene paralogs to be identical should be shown.

The answer regarding how much difference it makes was described in the manuscript ("*Light chain coherence is still visible even if light chain V gene paralogs are not treated as the same gene, with light chain coherence of 64% for memory cells (Extended Figure 3).*"). [now **Extended Table 4**]

These paralogs are in a duplication discovered in:

Pech M et al. 1985. A large section of the gene locus encoding human immunoglobulin variable regions of the kappa type is duplicated. J Mol Biol. 183.

We have added this as a citation. There is a large literature on this, which we don't cite, but this article:

Lefranc M-P. 2001. Nomenclature of the Human Immunoglobulin Kappa (IGK) Genes. Exp Clin Immunogenet. 18.

has a diagram that shows the parallel gene layout.

Line 126. Are ambiguous V gene assignments counted as similar or different? If ambiguous V genes count as different, this could unintentionally include sequences from the same clonotype.

We always assigned a V gene, however as discussed above, for **Figure 2** we now require the **same** V gene, and positive evidence that cells come from different clonotypes, so V gene uncertainty should no longer be an issue.

Line 159. Typo: "and if Y is similar to Z, they both go in the same group, and therefore X and Z must go in the same group..."

Thank you, this text has been replaced.

Line 163-169. Are these initial groupings defined using only the heavy chain? This should be specified. The results of this analysis are inadequately detailed in the text. For instance, what is the light chain coherence for transitive clonotypes? We are only told that it is lower than in Figs. 1-2.

We agree with the referee that our previous text did not adequately explain this process. We have provided a complete example in the main body of the paper as part of our improvements to this section. The new text is as follows:

“We provide a concrete example at 90% identity for the sake of clarity. We consider only memory cells and only computed clonotypes consisting entirely of memory cells. We define groups by first defining when two cells are similar. We call two cells similar if they belong to the same clonotype or if they have identical heavy chain V genes and 90% identical CDRH3 amino acid sequences. We then place two cells X and Y in the same group if there is a sequence of cells $X = X_1, \dots, X_n = Y$ such that for each i , X_i is similar to X_{i+1} . The multiple “hops” between these two cells make them transitively similar. This process is a well-known mathematical notion, which we call transitive grouping. It places every cell in a group and yields non-overlapping groups. We analyzed these groups for light chain coherence by examining pairs of cells from different donors within the same group.”

We have also clarified in the text the relationship between light chain coherence in transitive clonotypes and the percent identity used to form transitive groups.

Lines 214-242. In this section, the authors appear to be testing their hypothesis that recurrent clones occur across donors purely by chance. They test this by performing simulations with OLGA. Their simulations showed significantly fewer recurrent clones than their empirical data, in opposition to their hypothesis. Rather than changing their conclusions, the following paragraph appears to imply that their hypothesis was correct but that the model used in simulations was improperly specified.

We ran new simulations using soNNia, with models trained on our data. The simulations now recapitulate our training data accurately, unlike baseline OLGA, and predict slightly more recurrences than we observe in our data. We note that this directly supports our hypothesis and thank the referee for the suggestion to use a simulator that can be easily parameterized using real data.

Line 231. What is the substitution rate, and how does it differ from somatic hypermutation level? If naive sequences are being simulated, why are there any substitutions at all?

That particular text is now gone, but the concept lives on in the current **Figure 3**. We added explanatory text in the main text and in the Figure legend. The substitutions are the substitutions visible in an alignment of an antibody junction region to concatenated reference sequences. These could be mutations arising from recombination or SHM; it is impossible to discern which of these applies to a given base from a given antibody based on antibody sequence alone.

Line 236. Is 17.8% the true value of simulated VDDJ junctions, or is that the value estimated from the analysis pipeline? It would need to be the latter to be directly comparable to the real naive antibodies. How are VDDJ junctions inferred and how accurate is that process?

Our code is the only publicly available code to annotate VDDJ junctions and is described in Jaffe *et al.* 2022 *bioRxiv*. After running new (and properly parameterized) simulations with soNNia, we see a comparable number of VDDJ junctions in the simulations compared to our data. We thank the referee for this feedback, as it led us to discover a bug in the analysis code that we ran on the simulated data. The findings of our paper are stronger and more compelling thanks to the referee. Manual inspection of VDDJ alignments for real data suggests that they are generally credible, however we are confident that D gene assignments for VDJ and VDDJ antibodies alike are in general wrong at a fairly high rate because we observe sporadically different assignments within clonotypes.

Lines 244-250. It is unclear what analysis was done here, what the results were, and what they mean in the context of the other analyses.

We refer to **Figure 3d** (formerly **Figure 4c**) and the text. We did this analysis because our work is about recurrent antibodies, which are biased towards less complex junctions, and we wondered if light chain coherence applies to antibodies with more complex junctions (as many do). The figure documents this exact behavior.

Fig 1: Can the authors speculate as to why the number of memory and naive cell pairings is roughly the same for each bin?

Good question. The short answer is that we don't know. The longer answer is that there is some balance between initial generation of sequences, and mutation/selection, and it's complicated. This question is also not directly aligned with the investigations of our work so we would prefer not to speculate in the text.

There does not appear to be any formal treatment of significance or uncertainty in the results presented. For example, the central claim of ~80% light chain coherence in memory B cells is presented as being surprising. However, the null hypothesis for why this result is noteworthy is never explicitly stated or calculated. If heavy and light chains are paired together randomly during initial V(D)J recombination, what level of coherence would be expected? At the very least, the results should be tested through a permutation test. Naive B cells are not an appropriate null hypothesis because the V gene usage of the memory B cell pool is expected to be far more biased (for example, <https://doi.org/10.1182/blood-2010-03-275859>). The authors should add a control line to Fig.1 that include randomly chosen cells with the same V/J gene and CDRH3 length distribution to match each of the points on the red lines. Additional biases are likely to be induced by the clonal relationships within each of the donors, as mentioned above.

We have done this, and show that the results of permutation are comparable to the light chain coherence seen at low percent identity thresholds and also in the majority of naive cell pair comparisons.

Remove all italics used for emphasis. All terms and analyses should be adequately specified to be understandable without italics.

We have done this.

Extended Figure 4 is a nice data visualization, but it is difficult to see how the percentages presented in the text relate to the figure.

We have improved the legend accompanying this figure to clarify this for readers. The underlined text below represents the addition to the existing legend:

“Extended Figure 2. Light chain coherence is visible by sequence similarity. Each point represents a pair of memory cells from different donors. Heavy and light chain edit distances are plotted, using the amino acids starting at the end of the leader and continuing through the last amino acid in the J segment. Points with identical coordinates are combined by showing a large point whose area is proportional to the number of such points. (a) Cell pairs are displayed if the two cells in the pair have the same CDRH3 amino acid sequence. To increase readability, only one third of such pairs were selected at random for display. Of the pairs, 78% have light chain edit distance ≤ 20 . This number (78%) is the fraction of cell pairs lying below the horizontal line at

light chain edit distance 20, and was computed separately. It is proportional to the fraction of red below the line, if overlap is taken into account. (b) [control] The same number of cell pairs were selected at random for display, without regard to CDRH3. Of the pairs, 9% have light chain edit distance ≤ 20 .

In Figures 3a and 3b, remove the 'a' symbol from the ggplot legend.

This is no longer necessary as we have dropped the COSUM analysis from the manuscript.

The referencing of the figures is inconsistent. It seems to shift from, for example, Figure 3(a) to Figure 4a depending on the figure.

We have now referenced all figures using the format “**Fig. 3a.**”

Lines 30-38: there are multiple claims made in this paragraph, none of which have citations to support them.

In an effort to provide readers with helpful material, we have included citations as to the potential future utility of structural modeling (Raybould 2021) and possible orthogonal data that could be used to assess the validity of computational clonotyping efforts (Miller Nat Biotechnol 2022).

While this is true, please provide a source to support your claim. [However, identifying genuine recurrence can be precarious. In principle one could base such an analysis on pairs of computed clonotypes, though an obvious concern would be that two computed clonotypes were in fact part of a single true clonotype that was incompletely grouped...]

Our opinion is that adding a source (or many sources?) would likely not be helpful to the reader, particularly since this work is not about clonotyping. . What we say is part of the assertion that “computational clonotyping is imperfect.”

Lines 131-132 why do you expect the true rate of coherence to be approximately equal?

The current text says that:

“Recurrences (separate recombination events) occur between different donors and also within single donors; from first principles light chain coherence should occur at a comparable rate in a single donor.”

We think that light chain coherence is a manifestation of the basic biology of B cells responding to an antigen. Certain heavy chains happen to work well, and for each of those only a few light chains provide the best binding. This is really a statement about single B cells, and so it should not matter which individual they reside in.

Line 236 (held within parentheses) please provide a reference that a VDDJ antibody would be far less likely to recur by chance.

In our new simulation data (using soNNia) models, we observe effectively no (mean across n=10 replicates = 1 recurrence) recurrent VDDJ sequences. VDDJs antibodies occupy a much larger space of possibilities and are inherently more complex sequences. This text is now irrelevant as our new simulation results recapitulate the

rarity of both VDDJ sequences and recurrences thereof. Similarly, our reference 31 highlights the rarity of these cells in the human antibody repertoire—inherently rare sequences are also inherently less likely to recur.

Lines 256-257: please provide a source to support this claim [In nature, many heavy chain configurations yield effective binding of a given antibody target.]

We have changed the text to say the following:

“In nature, many heavy chain configurations yield effective binding of a given antigen.”

Lines 308-309: please provide a source to support this claim. [By virtue of how V(D)J recombination works, the light chain sequences of antibodies carry less information than the heavy chain sequences.]

We have added a citation.

The authors introduce “transitive” clonal grouping, which does not appear any different from single linkage hierarchical clustering, which is commonly practiced in the field to identify clonal groups.

We thank the referee for asking this question. We have clarified the text and hope that it is now self-evident that we did not do single-linkage hierarchical clustering. The Wikipedia article on that topic refers to completely different algorithmic steps that we do not mention and did not perform. The transitive grouping that we use is not something we invented. It is a common mathematical notion, and it is just the simplest possible way of forming groups. In the manuscript we explain exactly what we do because we do not have a good reference and wish to maximize accessibility. We could give a pure mathematics reference but it would be incomprehensible to biologists and not particularly helpful to anyone.

The abstract appropriately summarizes the key findings from Figs. 1 and 2. However, the claim that “naive antibodies recur by chance” does not appear to be supported within the text, and in fact appears to be undermined by the simulation analyses in Lines 214-224 (see comments above).

As addressed in our response to the comments above, this statement no longer applies to our findings. We show that antibody recurrence is in fact predicted by simulations, and that such simulations recapitulate the statistical properties of natural repertoires sufficiently well for the purpose of this work.

The results section concludes with the sentence: “Indeed, our findings imply that all antibodies are recurrent, but at varying rates depending on their junction complexity and the prevalence of their cognate antigen.” This is not a meaningful statement, considering that rates of recurrence could vary between 0 and 100% percent.

We disagree that this sentence is not meaningful. In fact, we intended for this statement to be read exactly as the referee re-stated—that recurrence of an individual sequence is in fact predicated unpredictably on the complexity of its junction sequence and the relevant antigen ecology.

Further, in the statement, “Our work reveals that the light chain of functional antibodies is highly constrained”, the authors appear to use the term “functional” to mean “not naive.” This is confusing,

since within B cell repertoire analysis, “functional” is typically used to mean BCRs that encode a functional protein. This should be clarified throughout.

We appreciate this comment from the referee. This is an example of an overloaded term used in the literature. “Clonotype” is another example. We think that in the context of this work, readers will understand what is meant by functional. We note that the majority of recent antibody literature that is readily available at the top of the Google search “functional antibody” supports our choice of phrasing, including:

- Ndungo & Pasetti. “**Functional antibodies** as immunological endpoints to evaluate protective immunity against Shigella”. *Hum Vaccin Immunother* (2020). DOI: 10.1080/21645515.2019.1640427
- Hamamichi et al. “Novel method for screening **functional antibody** with comprehensive analysis of its immunoliposome”. *Sci Rep* (2021). DOI: 10.1038/s41598-021-84043-w
- Fendler et al. “**Functional antibody** and T cell immunity following SARS-CoV-2 infection, including by variants of concern, in patients with cancer: the CAPTURE study”. *Nat Cancer* (2021). DOI: 10.1038/s43018-021-00275-9
- Lofano et al. “B cells and **functional antibody** responses to combat influenza”. *Front Immunol* (2015). DOI: 10.3389/fimmu.2015.00336
- Depetris et al. “**Functional antibody** characterization via direct structural analysis and information-driven protein–protein docking”. *Proteins* (2021). DOI: 10.1002/prot.26280
- Pullen et al. “Selective **functional antibody** transfer into the breastmilk after SARS-CoV-2 infection”. *Cell Reports* (2021). DOI: 10.1016/j.celrep.2021.109959
- Li et al. “A **functional antibody** cross-reactive to both human and murine cytotoxic T-lymphocyte-associated protein 4 via binding to an N-glycosylation epitope”. *mAbs* (2020). DOI: 10.1080/19420862.2020.1725365
- Walsh et al. “A general approach for the site-selective modification of native proteins, enabling the generation of stable and **functional antibody**–drug conjugates”. *Chem Sci* (2019). DOI: 10.1039/C8SC04645J

Referee 2 comments

In this excellent study, the authors performed an extremely deep sequencing of the circulating human B cell repertoire, producing a dataset of ~1.6M natively paired BCR sequences. Large sequencing sets have been collected previously but not retaining native pairing. In addition to generating (and already making publicly available!!), an unprecedented dataset, the authors report the existence of light chain coherence in public, antigen-experienced BCRs. This is a fundamental immunological insight that could only be observed by examining a dataset of this size.

The manuscript is clear and well written, although a few areas could be a bit more concise. Also, the clarity of the data could be improved by converting a few tables to figures.

We have converted several tables to Figures and moved others to the Extended and Supplementary sections. Where possible, we have taken steps to simplify and shorten the manuscript while accommodating other referee requests.

For example, plotting the data in Figure 4e would make the differences between naive and simulated repertoires much easier to distinguish. Of course, the authors could retain the tables as supplementary material so that the quantitative data is still accessible if desired.

We thank the referee for this feedback and have moved much of the table-based data to Supplementary Tables to improve the readability of the manuscript.

The authors use a novel method for allele inference, however there is some confusion about the methodology. First, after performing a V-gene pileup, the authors identify positions of potential allelic variance (>4 occurrences and >=25% of the total pileup) and then mention that “each cell has a footprint relative to these positions”. It is not clear what the authors mean by footprint — are residues surrounding the potentially variant position somehow also being considered variants if they are also non-reference?

We have clarified the text with the following additions (shown in underlined text):

“Next, for each position along the V gene, excluding the last 15 bases (to avoid the junction region), we determine the distribution of bases that occur within these selected cells. We only consider those positions where a non-reference base occurs at least four times and represents at least 25% of the total. Then each cell has a footprint relative to these positions, which is its list of base calls for the given positions; we require that these footprints satisfy similar evidence criteria. Each such non-reference footprint then defines an “alternate allele”.”

We also added a sentence pointing to the section on donor reference analysis in our reference 47 (Jaffe et al. 2022 *bioRxiv*), which we think would be helpful to the reader. Please see our response to the following question for additional details.

Second, did the authors benchmark their allelic inference method against other inference methods? If not, such a comparison would seem appropriate—not to make any claims about the accuracy of any of the methods, but merely to give readers (who may be more familiar/comfortable with existing tools) a sense of reference. This is particularly relevant because the computational splitting of cells into naive and memory subsets uses a threshold of zero mutations for naive B cells, which would be highly sensitive to differences in allele inference.

A comparison of allelic inference methods could be helpful for some readers, and we have clarified our methods text appropriately to assist readers' understanding of our methods. Our allele reference methods are described in substantial detail in Jaffe et al. 2022 *bioRxiv*, including a worked example of *IGHV1-18*, which we hope will give readers a frame of reference as the referee suggests. However, we suggest that this work is not the appropriate place for an allelic inference comparative analysis, for the following reasons:

- It is true that the computational splitting would be sensitive to allelic inference flaws. However, we already know that some cells are misclassified, because our method (and other methods) cannot identify SHM in the junction region. Similarly, if usage of a V gene is sufficiently rare, our method would not call an allele even if present, and so we could misclassify cells using that gene. Other methods might or might not make such a call, but there would be little or no basis for knowing if the call was correct.
- It is unlikely that misclassification of a small fraction of cells would significantly affect our claims about light chain coherence. To that end, we did an experiment and added the following results in the main text:

“We tested the effect of antibody sequence misclassification by randomly swapping memory and naive labels for 10% of cells, finding still 82% coherence for memory cells but 17% (cf. 10%) coherence for naive, suggesting that naive coherence might be exaggerated by actual errors.”

- We think it would be unproductive to engage in an analysis without the correct data types in hand, so that results can be properly assessed (cf. first point). The right way to do the experiment is to obtain two data types from one individual: (1) their complete diploid genome sequence (accurate and complete at least on BCR regions); (2) deep VDJ data from single cells. This would be feasible as part of a separate investigation, and would facilitate a meaningful analysis that is not otherwise possible.

In response to a related question, we also added the following text to the Discussion section of the manuscript: “The immunoglobulin loci and their products are complex and challenging to analyze. It is reasonable to assume that some sequences treated as naive in this study are in fact antigen-specific as others have described in SARS-CoV-2 infection [ref]. Conversely, though progress has been made in the identification of novel germline alleles [refs], some sequences treated as memory in this study may in fact be naive and produced by novel alleles not detected by our inference methods.”

In Figure 4d (and the associated text) the authors mention “junction substitution rate” when discussing junctions with no N-addition at VD or DJ junctions. It is not clear what the authors mean by substitution rate, as these antibodies are all naive and would not be expected to contain SHM-induced substitutions. When performing sequence simulation with OLGA, it appears the authors used the default simulation settings, which were modeled on non-productive recombinations to isolate the specific likelihood of VDJ recombination events and minimize skewing caused by negative selection during early B cell development. As such, it is not necessarily surprising that the synthetic repertoires are somewhat distinct from observed naive repertoires. It may be useful to clarify this, which would strengthen the authors conclusion — if negative selection does indeed “sieve” the repertoire by removing unacceptable recombinations, it is likely that the sieved repertoires would be more similar across individuals than would be expected from coincident recombinations alone.

The first referee also had questions about the simulation results. We re-ran simulations using soNNia and find that the discrepancies are largely resolved, other than the simulations predicting a comparable and slightly larger number of recurrences than we observe in our data, which supports our hypothesis and this statement from referee 2. We also provide evidence of this “sieving” effect using simulation; we show these results in **Supplementary Table 3** below:

Data source	Simulation parameters		Average junction property						
	Unique sequences	Selection model	VDJ recurrences (# ± SEM)	VDDJ recurrences (# ± SEM)	Insertion length (NT)	Substitution rate	Substitution rate (cells w/ 0 insertions)	VDDJ rate	CDRH3 length (AA)
Real	--	--	754.0 (N/A)	0.0 (N/A)	5.0	15.6%	7.2%	0.52%	18.4
Naive	No	Post	1189.6 (±10.8)	1.8 (±0.7)	5.3	16.5%	7.3%	0.32%	18.1
Naive	Yes	Post	1217.3 (±16.9)	0.6 (±0.3)	5.4	16.6%	7.3%	0.32%	18.1
Memory	No	Post	1444.5 (±13.8)	1.0 (±0.3)	4.8	17.4%	7.7%	0.35%	17.4
Memory	Yes	Post	1632.0 (±14.8)	1.6 (±0.6)	4.6	17.5%	7.7%	0.28%	17.2
Naive	No	Pre	167.4 (± 5.8)	1.4 (±0.6)	8.7	16.7%	8.3%	1.32%	21.0
Naive	Yes	Pre	172.1 (± 3.6)	1.0 (±0.4)	8.8	16.7%	8.8%	1.30%	21.0
Memory	No	Pre	173.3 (± 6.1)	1.0 (±0.3)	8.8	16.8%	8.3%	1.34%	21.0
Memory	Yes	Pre	175.4 (± 4.5)	2.0 (±0.6)	8.7	16.7%	8.3%	1.35%	21.1

Referee 3 comments

Jaffe and colleagues described an analysis of single cell sequencing data of human antibody repertoires and addressed a long-standing question about specifics of heavy and light chain pairings. The textbook description of antibody repertoire generation tells that heavy and light chain pairing increases the diversity of antibodies by several orders of magnitude. The authors update this statement and show that while the naive diversity of heavy and light chain pairs is high, it is significantly reduced in mature B cells. This finding is extremely important for understanding features of functional antibodies and has applications in vaccine and drug design.

I read the manuscript with great interest. It is very well written, the conclusions are clear and justified. The cited papers are appropriate. I have several clarifying questions about the performed methods and experiments. I agree with the general layout, but I would be grateful for additional details - please see questions 1–4. I also think that additional computational experiments described in questions 4–7 will be interesting for immunologists.

We thank the referee for requesting these details and additional experiments, which we provide detailed responses to and information about below.

While it's known that flow cytometry can create both false positives and false negatives, the computational method described in lines 72–84 also can introduce artificial biases. E.g., V(D)J sequences that were classified as naive by flow cytometry can contain mutations wrt germline genes because they were derived from unknown alleles of germline genes. Similarly, V(D)J sequences that were classified as memory can be mutation-free because corresponding germline genes do not need mutations to recognize certain types of antigens. E.g., anti-SARS-CoV-2 antibodies reported by Kim et al., 2021 (<https://doi.org/10.1126/scitranslmed.abd6990>) have these features. While the frequency of this phenomena has not been studied, it's important to take it into account. I would recommend adding a remark on possible computational biases and comment on their impact on the obtained results.

We thank the referee for this comment and agree this is a helpful clarification. We have added the following sentences in the Discussion section:

“The immunoglobulin loci and their products are complex and challenging to analyze. It is reasonable to assume that some sequences treated as naive in this study are in fact antigen-specific as others have described in SARS-CoV-2 infection [ref]. Conversely, though progress has been made in the identification of novel germline alleles [ref], some sequences treated as memory in this study may in fact be naive and produced by novel alleles not detected by our inference methods.”

Lines 60–62 “We exclude related cells (i.e., those in the same clonotype) because they use the same VDJ genes by definition” and following them lines 88–90: “We only considered pairs of cells with the same heavy chain V gene and the same CDRH3 length, and whose cells came from different donors”. Could authors clarify how clonal expansion has been taken into account? Does the sentence in lines 60–62 mean that authors keep only one cell from each clonotype? Let's assume that Donor 1 and Donor 2 have clonal lineages with similar heavy chains formed M and N cells, respectively. Does it mean that authors analyze MxN pairs of cells or only one pair? If MxN pairs are considered, then they can introduce artificial biases caused by overrepresentation of pairs of cells with similar chains.

We thank referee 3 for this comment and note that referee 1 had a similar question. We performed separate computational experiments using one representative cell per clonotype per pair (i.e. only one pair, not MxN pairs), which are now part of the main Figures.

A follow-up question: Let's again assume that Donor 1 and Donor 2 have clonal lineages with similar heavy chains formed M and N cells, respectively. What is the coherence inside clonal lineages? Is it 100%? Do authors observe dual light chains (doi:10.1016/j.coi.2014.01.012)?

We thank the referee for this question. As the referee implied, within typical clonal lineages we expect 100% light chain coherence as all cells within a lineage share an ancestral cell with a defined rearrangement and thus by definition are coherent. The results in our paper deliberately utilize “canonical” rearrangements with one heavy chain and one light chain—we do not include three-chain clonotypes (HLL, HHL).

With regards to allelic inclusion, we observe cases of clonotypes having three chains (HLL), where the two light chains use the same V gene. For example there is a clonotype having 40 cells which use V genes *IGHV3-23*, *IGKV3-20*, *IGKV3-20*. The light chain CDR3s for the two light chains in this case are highly similar (one typical cell shown):

**CQQYGSSPPWTF
CQQYGSS-SWTF**

Similarly, there is a clonotype having 9 cells with V genes *IGHV3-73*, *IGKV2D-29*, *IGKV2D-29*.

So possibly there are two interchangeably functional light chain V genes in such cases, but asserting this would be a stretch. We note also that single-cell clonotypes with three chains are likely enriched for doublets; we would not attempt to infer biology from these.

Lines 90–91: “We divided the pairs into eleven sets by their CDRH3 amino acid percent identity, rounded down to the nearest 10%”. Could authors describe how large these clusters are and how many donors each cluster represents?

We have added this information to the Figures and also in the relevant Extended and Supplementary Tables; referee 1 also had the same question. We thank the referees for requesting these numbers.

Could authors add an analysis of heavy and light V gene pairings? I think a heatmap showing IGHV genes as rows and light chain V genes as columns (a heatmap cell = the fraction of clonotypes with a pair of V genes) would be illustrative. Which IGHV have strong preferences toward light chain V genes and which are promiscuous? Such analysis can be useful for antibody drug design.

We thank the referee for this suggestion. Our paper is focused on observing light chain coherence and determinism, and we do so by focusing on individual junction/CDR3 rearrangements. We believe that figures such as this are prone to misinterpretation and are confounded by donor and naive/memory effects. To that end, we have made **Supplementary Figure 1**, and welcome any additional feedback. The first two panels show (a) superfamily-level and (b) gene-level associations between heavy chain V genes and light chain V genes. The second two panels show (c) the results of a Monte Carlo-bootstrapped chi-squared test and its residuals (rainbow colormap) to show the directionality of association between individual heavy chain V genes and light chain V genes and (d) the degree of contribution of each V_H/V_L gene combination to the chi-squared test.

A similar analysis of CDRH3 lengths and light chain V genes could be also very useful.

We agree with the referee and have made **Supplementary Figure 2** accordingly.

Do existing antibody structures confirm findings described by authors?

Human antibody structures in PDB do not come with the necessary information to answer this question—namely which human an antibody came from, which germline genes are used by an antibody, or which antibodies are clonally related. There are <1,000 unique human antibody sequences in PDB, which would not convincingly confirm our findings. However, we have added 280,669 additional cells' worth of data from Phad et al. 2022 *Nat. Immunol.* in addition to previously published datasets from the literature, all of which confirm our findings in relation to light chain coherence.

Lines 51–52: “four heavy chain V genes and two light chain V genes” - please list those genes.

Thank you for this comment; we have listed these genes (*IGHV3-23*, *IGHV3-30*, *IGHV3-30-3*, *IGHV3-48* and *IGKV3-11* and *IGKV3-15*) in the text.

Reviewer Reports on the First Revision:

Referees' comments:

Referee #2 (Remarks to the Author):

The authors have responded positively to my comments.

Referee #3 (Remarks to the Author):

A. In the revised work, Jaffe and colleagues described light chain coherence, a phenomenon of preferential pairings between heavy chains and light chains in antibodies produced by non-naive B cells. The authors used newly generated and publicly available single-cell data to support their findings.

B. The reported findings are novel and valuable for a wide range of disciplines: from disease studies to rational design of vaccines.

C. The authors have addressed reviewers' comments including important concerns raised by Reviewers 1 and 3 and performed additional computations of coherence % by collapsing clonally related sequencing and using one sequence per clonotype. Even though the resulting coherence values turned out to be slightly lower as compared to uncollapsed results, I am still convinced that the described coherence effect takes place. I thus would like to thank authors for adding these additional results.

D. The main text now includes permutation tests as Reviewer 1 suggested. I would recommend to (i) provide the p-value computed by the permutation test and (ii) add a description of error bars for the gray permutation test line in Figure 1. Statistical tests described in Supplementary Figure 1 are appropriate.

E. The conclusions are supported by the text.

F. Please take a look Figure 3a - the second line showing the alignment seems off - probably a non-monospace font was used there.

G.

>appropriate credit to previous work?

Yes.

H. I recommend including a statement about the impact of clonality to the abstract and providing coherence % for clonal collapsed data.

Author Rebuttals to First Revision:

Summary

We again thank the referees for their comments and feedback.

We have made the following revisions in response to comments from reviewer 3:

- a. *D. The main text now includes permutation tests as Reviewer 1 suggested. I would recommend to (i) provide the p-value computed by the permutation test and (ii) add a description of error bars for the gray permutation test line in Figure 1. Statistical tests described in Supplementary Figure 1 are appropriate.*

We have clarified the legend of Figure 1 and methods text accordingly.

- b. *F. Please take a look Figure 3a - the second line showing the alignment seems off - probably a non-monospace font was used there.*

Yes thank you, we accidentally dropped the font. This is now fixed.

- c. *H. I recommend including a statement about the impact of clonality to the abstract and providing coherence % for clonal collapsed data.*

Using the data of this work, we found 82% light chain coherence when all cells were used, as compared to 79% when cells were clonally collapsed. Both of these numbers are ~80%, and we were therefore not sure that it would be helpful to exhibit the small difference in the abstract. Instead we have added text, highlighted in red:

for memory (functional) antibodies it is ~80%, even if only one cell per clonotype is used

We also made one additional very small change, highlighted in red:

all but three cells use the light chain V gene *IGKV2-30*

to

all but four cells use the light chain genes *IGKV2-30* and *IGKJ2*,

as this paints a more complete picture of the example.